# CODEI/O: Condensing Reasoning Patterns via Code Input-Output Prediction

Junlong Li [1 2 3 *]   Daya Guo [1]   Dejian Yang [1]   Runxin Xu [1]   Yu Wu [1]   Junxian He [3]

## Abstract

Reasoning is a fundamental capability of Large Language Models. While prior research predominantly focuses on enhancing narrow skills like math or code generation, improving performance on many other reasoning tasks remains challenging due to sparse and fragmented training data. To address this issue, we propose CODEI/O, a novel approach that systematically condenses diverse reasoning patterns inherently embedded in contextually-grounded codes, through transforming the original code into a code input-output prediction format. By training models to predict inputs/outputs given code and test cases entirely in natural language as Chain-of-Thought (CoT) rationales, we expose them to universal reasoning primitives—like logic flow planning, state-space searching, decision tree traversal, and modular decomposition—while decoupling structured reasoning from code-specific syntax and preserving procedural rigor. Experimental results demonstrate CODEI/O leads to consistent improvements across symbolic, scientific, logic, math & numerical, and commonsense reasoning tasks. By matching the existing ground-truth outputs or re-executing the code with predicted inputs, we can verify each prediction and further enhance the CoTs through multi-turn revision, resulting in CODEI/O++ and achieving higher performance. Our data and models are available at https://github.com/hkust-nlp/CodeIO.

## 1. Introduction

Reasoning is a fundamental aspect of human cognition and problem-solving, forming the basis for quickly transferring and adapting to new tasks (Dehaene et al., 2004; Knauff

& Wolf, 2010; Wang & Chiew, 2010). It is also recognized as a cornerstone of advanced Large Language Models (LLMs) and a critical step toward achieving Artificial General Intelligence (AGI) (Huang & Chang, 2022; Qiao et al., 2022; Jaech et al., 2024; Xiang et al., 2025). Current approaches, however, face a fundamental paradox: while tasks like math problem solving (Shao et al., 2024; Yang et al., 2024; Zeng et al., 2024; Ying et al., 2024; Toshniwal et al., 2024) and code generation (Roziere et al., 2023; Mistral-AI, 2024; Zhu et al., 2024; Hui et al., 2024) benefit from abundant structured training data, most other reasoning domains—including logical deduction, scientific inference, and symbolic reasoning—suffer from sparse and fragmented supervision signals. As a result, it becomes crucial to identify training data that is rich in diverse reasoning patterns while remaining scalable to obtain.

We believe that real-world code programs reflect the integration of a wide range of reasoning patterns across diverse contexts, making them an ideal source for training while minimizing the risk of overfitting. However, conventional continual pre-training on raw code is suboptimal because the relevant reasoning signals are often implicit and intertwined with noisy information. Even the cleaner objective of directly training on text-to-code generation also faces challenges, as it is constrained by the requirement to generate code-specific syntax, making it difficult to generalize to tasks beyond code-specific ones. To address such limitations, we propose transforming raw code files into executable functions and designing a more straightforward task: given a function along with its corresponding textual query, the model needs to predict either the execution outputs given inputs or feasible inputs given outputs entirely in natural language as CoT rationales. This approach aims to disentangle core reasoning flow from code-specific syntax while preserving logical rigor. By gathering and transforming functions from diverse sources, the resulting data incorporates a variety of foundational reasoning skills, such as logic flow orchestration, state-space exploration, recursive decomposition, and decision-making. Learning from these samples across the diverse contexts provided by the raw code files enables models to gain repeated exposure to these reasoning processes, allowing them to better internalize these skills.

Similar to continual pre-training on raw code, our code input/output prediction learning is introduced as a distinct

*Work done during internship at DeepSeek-AI. [1]DeepSeek-AI [2]Shanghai Jiao Tong University [3]HKUST. Correspondence to: Junlong Li <lockonlvange@gmail.com>, Junxian He <junxianh@cse.ust.hk>.

*Proceedings of the 42nd International Conference on Machine Learning*, Vancouver, Canada. PMLR 267, 2025. Copyright 2025 by the author(s).

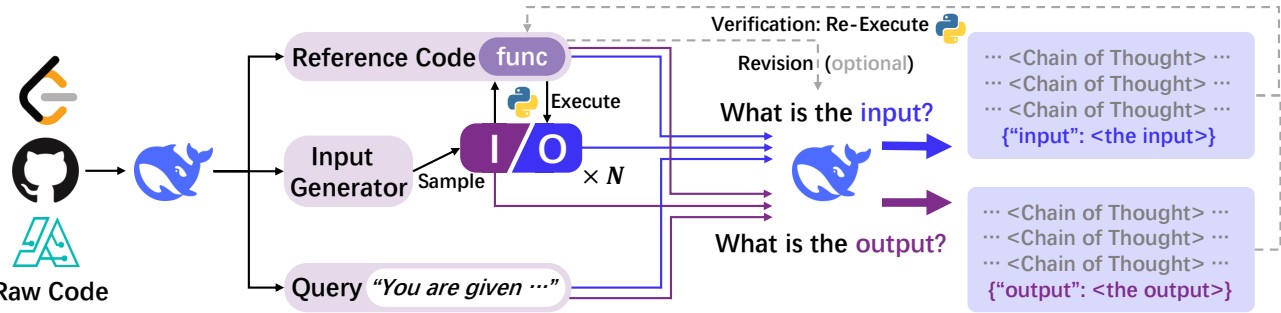

Figure 1: Overview of our training data construction: Raw code files are gathered from various sources and converted into a unified format. Input-output pairs are then generated by executing the code, while natural language CoTs for predictions are collected from DeepSeek-V2.5. The verified CoTs can undergo optional revisions to further enhance reasoning chains.

training stage positioned before general instruction tuning in a two-stage manner, serving as an intermediate step to enhance the reasoning abilities of the base model. The prompt includes the function, the textual query, and the given input or output, while the response is directly collected by prompting a strong open-source model, DeepSeek-V2.5 (DeepSeek-AI et al., 2024). Notably, the instances for input-output prediction are highly scalable to collect, as we can sample hundreds of inputs from a separate Python input generator for each function and execute the code to obtain ground-truth outputs. Finally, we collect over 450K functions from multiple sources, and for each function, several input-output pairs are generated by executing the corresponding code. Synthesizing CoTs for them results in a total of 3.5M training samples, yielding the CODEI/O data. To further leverage the verifiable characteristics of code, we verify all predictions based on code execution and prompt DeepSeek-V2.5 for a second turn of revisions on the responses it initially got wrong. These multi-turn revisions are then concatenated into longer responses. The resulting CODEI/O++ dataset further enhances model performance, demonstrating the effectiveness of this refinement process.

We validate the effectiveness of CODEI/O and CODEI/O++ across four base models with parameter sizes ranging from 7B to 30B . Assessments across 14 different benchmarks show training on them enhances performance on a diverse range of reasoning tasks, not only limited to code-related tasks but also more generalized tasks such as logic, symbolic, mathematical & numerical, scientific, commonsense, etc. Compared to several strong data baselines, such as OpenMathInstruct2 (Toshniwal et al., 2024), OpenCoder-SFT-Stage1 (Huang et al., 2024), WebInstruct (Yue et al., 2024), and high-quality raw code (Ben Allal et al., 2024), CODEI/O achieves not only higher average scores across all four tested base models but also more balanced performance – Instead of boosting scores on only a small subset of evaluation benchmarks while causing declines on others, CODEI/O delivers consistent improvements across nearly

all benchmarks, demonstrating balanced and generalizable reasoning abilities.

## 2. CODEI/O

Our data construction pipeline is presented in this section. We begin with collecting raw code files from various sources (§2.1). They are then transformed into a unified format (§2.2). Next, I/O pairs are sampled from the transformed functions (§2.3). Finally, the complete training dataset is assembled (§2.4). An overview is depicted in Figure 1.

### 2.1. Collecting Raw Code Files

The effectiveness of CODEI/O lies in selecting diverse raw code sources that encompass a wide range of reasoning patterns. To achieve this, we select sources with different emphases: **CodeMix**, a large collection of raw Python code files retrieved from an in-house code pre-training corpus, where we filter out files that are either overly simplistic or excessively complex; and **PyEdu-R** (reasoning), a subset of Python-Edu (Ben Allal et al., 2024) that focuses on complex reasoning tasks such as STEM, system modeling or logic puzzles. To avoid overlap with CodeMix, we deliberately exclude files centered on pure algorithms. Beyond these two sources, we also incorporate high-quality code files from a variety of smaller, reputable sources, including comprehensive algorithm repositories, challenging math problems, and well-known online coding platforms. In total, merging these sources yields approximately 810.5K code files. Further details on the data sources can be found in Appendix C.1.

### 2.2. Transforming to a Unified Format

The collected raw code files often lack structure, contain irrelevant elements, and are hard to execute in a self-contained way. Therefore, we preprocess them using DeepSeek-V2.5 (DeepSeek-AI et al., 2024), which refines them into a unified format that emphasizes main logical functionality and

```
def change_ref(amt, coins):
    if amt <= 0: return 0
    if amt != 0 and not coins: return float("inf")
    elif coins[0] > amt: return change_ref(amt,
coins[1:])
    else:
        use_it = 1 + change_ref(amt - coins[0],
coins)
        lose_it = change_ref(amt, coins[1:])
        return min(use_it, lose_it)
```

**Query**

You are given an amount of money `amt` and a list of coin denominations `coins`. Your task is to determine the minimum number of coins needed to make up the amount `amt` using the given denominations. If it is not possible to make the amount with the given coins, the function should return `inf` (infinity).

**Given input = {"amt": 25, "coins": [1,4,7]}, predict output**

1. Greedy Approach: We want to use as many of the largest denomination coins as possible to minimize the number of coins. However, the greedy approach doesn't always work for all coin systems, so we need to consider all possible combinations.

2. Possible Combinations:
  - Using the largest coin `7`:
    - We can use `7` coins up to `3` times (since `3 * 7 = 21`).
    - After using `3` coins of `7`, we have `25 - 21 = 4` left.
    - To make `4`, we can use `1` coin of `4`.
    - Total coins used: `3 (for 7) + 1 (for 4) = 4` coins.

  - Using fewer `7` coins:
    - If we use `2` coins of `7`, we have `25 - 14 = 11` left.
    - To make `11`, we can use `2` coins of `4` (totaling `8`) and `3` coins of `1`.
    - Total coins used: `2 (for 7) + 2 (for 4) + 3 (for 1) = 7` coins.

  - Using `1` coin of `7`:
    - We have `25 - 7 = 18` left.
    - To make `18`, we can use `4` coins of `4` (totaling `16`) and `2` coins of `1`.
    - Total coins used: `1 (for 7) + 4 (for 4) + 2 (for 1) = 7` coins.

  - Using no `7` coins:
    - We can use `6` coins of `4` (totaling `24`) and `1` coin of `1`.
    - Total coins used: `6 (for 4) + 1 (for 1) = 7` coins.

3. Optimal Solution: The combination that uses the fewest coins is the one where we use `3` coins of `7` and `1` coin of `4`, totaling `4` coins.
{"output": 4}

**Given output = 4, predict input**

1. Objective: We need to find an input where the minimum number of coins required to make up the amount `amt` is `4`.
2. Coin Denominations: The list of coin denominations should be such that the combination of coins that sums up to `amt` requires exactly `4` coins.
3. Example: Suppose we have coin denominations `[1, 2, 5]` and we want to make up the amount `8`. The optimal way to make up `8` using these coins is:
  - 1 coin of `5`
  - 1 coin of `2`
  - 1 coin of `1`
  - This sums up to `8` using `3` coins, which is less than `4`. So, this is not the correct input.
4. Another Example: Let's consider coin denominations `[1, 3, 4]` and the amount `6`. The optimal way to make up `6` using these coins is:
  - 1 coin of `4`
  - 2 coins of `1`
  - This sums up to `6` using `3` coins, which is still less than `4`.
5. Correct Example: Now, consider coin denominations `[1, 3, 4]` and the amount `8`. The optimal way to make up `8` using these coins is:
  - 2 coins of `4`
  - This sums up to `8` using `2` coins, which is less than `4`.
6. Final Example: Let's try coin denominations `[1, 2, 5]` and the amount `13`. The optimal way to make up `13` using these coins is:
  - 2 coins of `5`
  - 1 coin of `2`
  - 1 coin of `1`
  - This sums up to `13` using `4` coins, which matches our desired output.
{"input": {"amt": 13, "coins": [1, 2, 5]}}

Figure 2: Two examples for the collected responses for input and output prediction respectively.

makes it executable for us to collect input-output pairs for later prediction tasks. This transformation organizes the data into the following components, and we provide a complete example in Table 10 in Appendix G: **1) Cleaned Reference Code:** We preprocess the raw code files by cleaning and refactoring the code to extract core logical functionalities into functions. Non-essential elements like visualization (e.g., print, plot) and file processing (e.g., read, write) are excluded. **2) Main Entrypoint Function:** A main entrypoint function is added to summarize the overall logic of the code. It can call other functions or import external libraries and must have non-empty arguments (inputs) as well as return meaningful outputs. All inputs and outputs are required to be JSON-serializable to facilitate further processing. **3) Input/Output Description:** The inputs and outputs of the main entrypoint function are clearly defined, including information on data types, constraints (e.g., output ranges), or more complex requirements (e.g., keys in a dictionary). **4) Input Generator:** Rather than generating test cases directly, a standalone rule-based python input generator function is created. This generator returns non-trivial inputs that follow the requirements of the main entrypoint function. Randomness is applied subject to constraints, enabling scalable data generation. **5) Query:** A concise problem statement is generated based on the main entrypoint function, serving as a

query to describe its intended functionality of the code.

### 2.3. Collecting Input and Output Pairs

After converting the collected raw code files into a unified format, we sample multiple inputs using the input generator for each function and obtain the corresponding outputs by executing the code. To ensure the outputs are deterministic, we skip all functions that include randomness, such as those using `import random`. During the execution of these codes, we also impose a series of limits on the runtime and the complexity of the input/output objects (details in Appendix A). For each transformed function, we sample multiple input-output pairs, with the exact number depending on the source from which it originates (details in Appendix C.2). After filtering out non-executable code, samples that exceed the runtime limit, and input-output pairs surpassing the desired complexity, we obtain 3.5M instances derived from 454.9K raw code files. The distribution of input and output prediction instances is roughly balanced at 50%/50%.

### 2.4. Building Samples for Input-Output Prediction

After collecting the input-output pairs as well as the transformed functions, we need to assemble them into a trainable format. For the supervised fine-tuning process we adopt, a

prompt and a response are needed for each training sample. Since we aim for the input-output prediction tasks, we construct the prompt using a designed template to combine the function, the query, the reference code, and either a specific input or output. We provide an example prompt in Figure 8 in Appendix G. The response should ideally be a natural language CoT to reason about how to derive the correct output or a feasible input. In general, we choose the following two ways to construct the desired CoT responses:

**Direct Prompting – CODEI/O** While having full executable code theoretically allows us to generate reliable execution trajectories as responses, two challenges arise: 1) Obtaining a deterministic reverse function for input prediction is impractical; 2) Automatically constructed trajectories are constrained by pre-designed templates and lack the expressiveness and generalizability of free-form natural language reasoning. Thus, we adopt a fully LLM-based approach for synthesizing all the desired responses using DeepSeek-V2.5, as it has top-tier performance but extremely low cost compared to other advanced LLMs. The dataset generated here is referred to as CODEI/O. We provide two examples of collected responses in Figure 2.

**Making Full Use of Code – CODEI/O++** A common approach to enhance data quality is reject sampling (Yuan et al., 2023), where incorrect predictions are discarded. Though this approach suits CODEI/O well as we can verify all responses by re-executing the codes, we find it leads to suboptimal performance (§4.1). Therefore, we take an alternative approach to fully utilize the execution feedback from our reference code. For responses with incorrect predictions, we append the feedback as the second turn of input messages and ask DeepSeek-V2.5 to regenerate another response. In practice, we capture multiple types of feedback: For output prediction, we simply inform the model that it generated an incorrect answer. For input prediction, we additionally provide the executed output based on the incorrect input. For instances where the code fails to execute (e.g., due to a format error, argument mismatch error, or other runtime error), we also include these feedback explicitly.

After the second turn, we re-check the correctness of the newly generated responses. We then construct the final response by concatenating all of the four components: Turn 1 response + Turn 1 feedback + Turn 2 response + Turn 2 feedback. For correct responses in the first turn, the Turn 1 feedback is simply "Success" with no Turn 2 contents. In general, in first turn, 50% of the responses are correct and 10% of the incorrect ones can be successfully revised in the second turn. Similar to CODEI/O, we keep all responses, either correct or incorrect, after the revision. The dataset we collect through this way is referred to as CODEI/O++, and we provide a complete example in Table 11 in Appendix G.

## 3. Experiments

### 3.1. Settings

**Models** We select the following base models as the backbones: Qwen 2.5 7B Coder (Hui et al., 2024), Deepseek v2 Lite Coder (MoE) (Zhu et al., 2024), LLaMA 3.1 8B (Dubey et al., 2024), and Gemma 2 27B (GemmaTeam et al., 2024). These models were chosen for being the most advanced base models currently, differing in architecture, size, and pre-training focus. Notably, we include two coder models, as previous studies have shown that coder models exhibit stronger reasoning capabilities compared to general-purpose models (Suzgun et al., 2023; Shao et al., 2024).

**Instruction Tuning Data** We utilize an in-house instruction-tuning dataset containing approximately 1.18M samples from different languages, encompassing a wide range of domains such as math, coding, writing, and more. Tuning the model on this dataset enables it to effectively follow diverse instructions, making it applicable to and testable on a broad spectrum of downstream tasks.

**Training Setups** Similar to continual pre-training, we employ a two-stage training strategy in most of our experiments. The first stage involves training on the CODEI/O or CODEI/O++ dataset, followed by a second stage of general instruction-tuning.

The reason for adopting this two-stage training approach is rooted in the characteristics of our datasets. The CODEI/O(++) dataset contains a significantly larger number of samples compared to the instruction-tuning data. Simply mixing the two datasets would result in a biased distribution, which could lead to insufficient learning on the instruction-tuning data. This might prevent the model from fully demonstrating its capacity to follow diverse instructions in downstream tasks. To address this, the two-stage training first strengthens the model as a more robust base model for general reasoning, and then adapts it into a versatile instruction-following model through instruction tuning. Detailed training hyper-parameters are in Appendix E.

**Evaluation Benchmarks** We evaluate all models on these benchmarks: DROP (Dua et al., 2019), WinoGrande (Sakaguchi et al., 2020), GSM8K (Cobbe et al., 2021), MATH (Hendrycks et al., 2021b), MMLU-STEM (Hendrycks et al., 2021a), BBH (Suzgun et al., 2023), GPQA (Rein et al., 2024), CruxEval (Gu et al., 2024), ZebraGrid (Lin et al., 2025). These benchmarks span multiple key reasoning domains, including science, math & numerical, symbolic, commonsense, logic, and code understanding. We also include two comprehensive benchmarks as well: LiveBench (White et al., 2024)[1], and KorBench (Ma et al., 2024). Besides

---

[1]We adopt the 2406-2407 split, excluding the code generation and instruction-following subtasks as they are not our focus.

Table 1: Main evaluation results on all benchmarks. WI = WebInstruct, OMI2 = OpenMathInstruct2, OC-SFT-1 = OpenCoder-SFT-Stage-1, PyEdu = PythonEdu. We also report the number of training samples for each dataset. Color-coded cells (green/red) are employed to denote improvements or declines relative to the single-stage baseline, with deeper shades indicating larger score shifts.

| 1st Stage Dataset | # (M) | Wino Grande | DROP | GSM 8K | MATH | GPQA | MMLU -STEM | LC -O | CRUX -I | CRUX -O | BBH -EN | BBH -ZH | Zebra Logic | Kor Bench | Live Bench | AVG |
|---|---|---|---|---|---|---|---|---|---|---|---|---|---|---|---|---|
| *Qwen 2.5 Coder 7B* | | | | | | | | | | | | | | | | |
| 2nd Stage Only | | 66.9 | 70.7 | 83.4 | 71.6 | 41.5 | 77.2 | 20.7 | 61.3 | 60.0 | 68.3 | 70.6 | 10.9 | 38.7 | 26.0 | 54.8 |
| WI | 3.5 | 66.3 | 73.5 | 87.0 | 71.4 | 39.1 | 77.5 | 18.3 | 59.1 | 61.6 | 68.6 | 68.7 | 10.2 | 42.5 | 26.0 | 55.0 |
| WI (Full) | 11.6 | 67.0 | 75.0 | 87.0 | 71.1 | 42.9 | 78.6 | 19.1 | 59.3 | 59.8 | 68.4 | 70.4 | 10.9 | 41.9 | 27.6 | 55.6 |
| OMI2 | 3.5 | 67.6 | 74.3 | 84.1 | 72.3 | 36.2 | 77.4 | 20.9 | 60.4 | 61.5 | 68.8 | 69.3 | 10.1 | 42.7 | 27.2 | 55.2 |
| OMI2 (Full) | 14.0 | 66.9 | 74.0 | 88.5 | 73.2 | 40.9 | 77.8 | 19.9 | 59.5 | 62.4 | 68.3 | 71.3 | 11.2 | 41.2 | 28.4 | 56.0 |
| OC-SFT-1 | 4.2 | 66.6 | 75.3 | 86.7 | 70.9 | 37.7 | 78.0 | 20.3 | 60.9 | 60.1 | 67.5 | 67.6 | 10.8 | 40.1 | 27.5 | 55.0 |
| PyEdu | 7.7 | 66.7 | 74.8 | 85.8 | 71.4 | 40.9 | 77.4 | 19.1 | 58.9 | 62.4 | 67.8 | 65.7 | 10.6 | 39.3 | 25.8 | 54.8 |
| CODEI/O | 3.5 | 67.9 | 76.4 | 86.4 | 71.9 | 43.3 | 77.3 | 23.7 | 63.6 | 64.9 | 69.3 | 72.8 | 10.7 | 44.3 | 28.5 | 57.2 |
| CODEI/O++ | 3.5 | 66.9 | 79.1 | 85.7 | 72.1 | 40.6 | 77.9 | 24.2 | 62.5 | 67.9 | 71.0 | 74.2 | 10.7 | 45.7 | 29.1 | 57.7 |
| *LLaMA 3.1 8B* | | | | | | | | | | | | | | | | |
| 2nd Stage Only | | 71.3 | 73.1 | 83.2 | 49.9 | 40.6 | 70.0 | 4.1 | 44.5 | 46.9 | 65.8 | 65.6 | 9.8 | 39.8 | 25.7 | 49.3 |
| WI | 3.5 | 72.1 | 76.3 | 82.8 | 52.8 | 42.9 | 69.6 | 4.1 | 44.0 | 44.8 | 64.5 | 67.8 | 10.0 | 42.7 | 23.1 | 49.8 |
| OMI2 | 3.5 | 72.2 | 74.8 | 86.2 | 58.9 | 38.2 | 70.1 | 5.8 | 46.1 | 46.4 | 67.4 | 68.6 | 9.5 | 40.3 | 24.5 | 50.6 |
| OC-SFT-1 | 4.2 | 71.0 | 71.9 | 81.8 | 51.1 | 38.2 | 68.4 | 5.7 | 43.5 | 44.9 | 65.6 | 67.6 | 10.5 | 42.0 | 24.7 | 49.1 |
| PyEdu | 7.7 | 70.6 | 69.6 | 83.2 | 49.8 | 42.4 | 69.1 | 5.2 | 43.1 | 44.5 | 64.0 | 65.6 | 10.2 | 42.6 | 25.7 | 49.0 |
| CODEI/O | 3.5 | 71.7 | 73.9 | 83.6 | 53.8 | 43.5 | 69.0 | 9.3 | 50.1 | 53.3 | 67.5 | 65.3 | 10.4 | 40.9 | 24.7 | 51.2 |
| CODEI/O++ | 3.5 | 71.8 | 75.1 | 84.0 | 53.2 | 40.9 | 68.4 | 10.0 | 50.4 | 53.1 | 70.0 | 70.6 | 10.5 | 43.2 | 28.1 | 52.1 |
| *DeepSeek Coder v2 Lite 16B* | | | | | | | | | | | | | | | | |
| 2nd Stage Only | | 68.4 | 73.4 | 82.5 | 60.0 | 38.6 | 68.5 | 14.8 | 53.0 | 54.9 | 61.1 | 69.2 | 6.7 | 44.7 | 26.6 | 51.6 |
| WI | 3.5 | 68.5 | 73.8 | 83.7 | 60.5 | 39.5 | 68.7 | 14.3 | 53.5 | 57.1 | 61.6 | 65.7 | 6.9 | 43.1 | 25.4 | 51.6 |
| OMI2 | 3.5 | 67.6 | 74.1 | 84.7 | 64.7 | 38.4 | 70.1 | 14.4 | 53.8 | 55.8 | 63.6 | 66.4 | 6.4 | 42.0 | 24.7 | 51.9 |
| OC-SFT-1 | 4.2 | 68.2 | 73.6 | 83.3 | 60.9 | 37.3 | 69.1 | 14.7 | 52.8 | 56.1 | 60.9 | 67.9 | 6.1 | 42.7 | 25.2 | 51.3 |
| PyEdu | 7.7 | 68.3 | 74.6 | 83.0 | 60.6 | 38.2 | 69.7 | 15.6 | 54.9 | 57.0 | 61.9 | 68.6 | 7.0 | 44.7 | 24.6 | 52.1 |
| CODEI/O | 3.5 | 68.4 | 74.6 | 83.6 | 60.9 | 38.6 | 70.3 | 18.7 | 58.4 | 62.8 | 63.1 | 70.8 | 7.8 | 46.0 | 26.1 | 53.6 |
| CODEI/O++ | 3.5 | 69.0 | 73.5 | 82.8 | 60.9 | 38.8 | 70.0 | 20.3 | 59.5 | 61.0 | 64.2 | 69.4 | 6.7 | 46.3 | 26.9 | 53.5 |
| *Gemma 2 27B* | | | | | | | | | | | | | | | | |
| 2nd Stage Only | | 72.4 | 80.1 | 90.1 | 66.3 | 44.4 | 82.8 | 19.1 | 62.5 | 66.9 | 77.1 | 80.4 | 13.5 | 47.8 | 30.0 | 59.5 |
| WI | 3.5 | 73.2 | 79.0 | 91.5 | 70.6 | 44.9 | 82.7 | 20.7 | 63.5 | 66.3 | 77.6 | 77.2 | 17.1 | 47.3 | 33.3 | 60.4 |
| OMI2 | 3.5 | 73.1 | 79.3 | 90.8 | 67.1 | 44.0 | 83.4 | 19.2 | 61.4 | 66.0 | 77.1 | 80.5 | 13.9 | 49.7 | 40.7 | 60.4 |
| OC-SFT-1 | 4.2 | 73.5 | 79.9 | 91.1 | 66.1 | 46.9 | 81.8 | 20.2 | 62.8 | 65.6 | 77.3 | 78.9 | 14.0 | 46.9 | 35.3 | 60.0 |
| PyEdu | 7.7 | 73.7 | 79.5 | 90.3 | 66.0 | 45.3 | 82.8 | 18.7 | 61.3 | 64.9 | 77.4 | 79.0 | 14.2 | 48.9 | 34.0 | 59.7 |
| CODEI/O | 3.5 | 75.9 | 80.7 | 91.2 | 67.4 | 44.9 | 83.3 | 22.4 | 65.0 | 70.3 | 77.9 | 78.7 | 14.6 | 49.1 | 31.3 | 60.9 |
| CODEI/O++ | 3.5 | 73.1 | 82.0 | 91.4 | 66.9 | 46.0 | 83.0 | 26.6 | 64.4 | 70.6 | 78.4 | 77.8 | 16.4 | 49.4 | 35.3 | 61.5 |

these established benchmarks, we test on two extra ones: BBH-ZH, a Chinese version of 9 BBH subtasks[2] as our instruction tuning data contains both English and Chinese examples, and LeetCode-O (LC-O), designed for bilingual output prediction for LeetCode questions with test cases. All evaluations are done with greedy decoding in a zero-shot setting, except for BBH-EN/-ZH where we use a 3-shot setup. Details of all benchmarks are in Appendix B.

**Baselines** The primary baseline is to directly fine-tune the base model on the instruction-tuning dataset in a single stage (2nd Stage only). This serves to evaluate whether the additional training stage provides any tangible benefits. We also select several strong datasets as baselines in the first Stage: *WebInstruct* (Yue et al., 2024): A large instruction-tuning dataset with 11.6M samples mined from the Internet and refined by LLMs. *OpenMathInstruct-2* (Toshniwal et al., 2024): A 14M-sample dataset focused on math problem solving, augmented from GSM8K and MATH using LLaMA 3.1 405B-Inst (Dubey et al., 2024). *OpenCoder-SFT-Stage-1* (Huang et al., 2024): A 4.2M QA-

---

[2]For clarity, BBH is referred to as BBH-EN in later sections.

Table 2: Key ablations we tested and the number of training samples under each condition. For a fairer comparison, we also provide results on a ~ 50% subset of CODEI/O to ensure the number of training samples remains comparable.

| | # (M) | Wino Grande | DROP | GSM 8K | MATH | GPQA | MMLU -STEM | LC -O | CRUX -I | CRUX -O | BBH -EN | BBH -ZH | Zebra Logic | Kor Bench | Live Bench | AVG |
|---|---|---|---|---|---|---|---|---|---|---|---|---|---|---|---|---|
| CODEI/O | 3.52 | 67.9 | 76.4 | 86.4 | 71.9 | 43.3 | 77.3 | 23.7 | 63.6 | 64.9 | 69.3 | 72.8 | 10.7 | 44.3 | 28.5 | 57.2 |
| ~ 50% subset | 1.59 | 67.5 | 74.7 | 86.7 | 71.6 | 42.9 | 77.3 | 23.0 | 62.8 | 65.9 | 69.1 | 70.8 | 10.5 | 42.1 | 28.9 | 56.7 |
| *Effect of prediction inputs or outputs only.* | | | | | | | | | | | | | | | | |
| I. Pred. only | 1.75 | 66.3 | 75.9 | 85.8 | 71.6 | 38.8 | 77.7 | 22.9 | 62.8 | 64.5 | 68.3 | 69.4 | 11.4 | 44.4 | 26.2 | 56.1 |
| O. Pred. only | 1.76 | 66.9 | 75.2 | 84.6 | 71.5 | 42.4 | 76.5 | 23.3 | 61.1 | 65.6 | 70.1 | 72.1 | 11.4 | 42.2 | 26.9 | 56.4 |
| *Effect of rejection sampling.* | | | | | | | | | | | | | | | | |
| w/o wrong | 1.79 | 66.8 | 74.9 | 87.4 | 71.5 | 39.1 | 76.7 | 22.6 | 62.6 | 66.6 | 68.3 | 71.9 | 11.5 | 42.6 | 27.8 | 56.5 |
| wrong→gt | 3.52 | 66.4 | 76.8 | 86.0 | 70.6 | 42.4 | 76.5 | 24.3 | 62.1 | 67.6 | 68.0 | 71.1 | 11.5 | 43.1 | 26.6 | 56.6 |

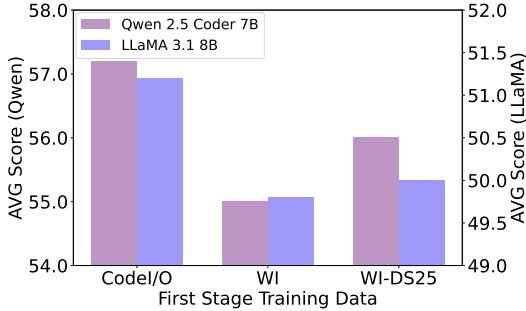

Figure 3: Average scores of Stage 1 training on CODEI/O, a 3.5M WebInstruct subset (WI) and an enhanced version distilled from DeepSeek-V2.5 Directly (WI-DS25).

pair dataset synthesized from general code data, covering diverse computer science domains. *Python-Edu* (Ben Allal et al., 2024): Following findings that continued pre-training on code tends to enhance reasoning, we adopt its full 7.7M code corpus and train on it using a standard language modeling loss. For *WebInstruct* and *OpenMathInstruct-2*, we use 3.5M subsets for most experiments to align with the size of our CODEI/O dataset, but also report the scores when training on the complete datasets for Qwen 2.5 7B Coder.

### 3.2. Main Results

We demostrate the main evaluation results in Table 1. As shown, CODEI/O provides universal gains across benchmarks, outperforming both the single-stage baseline and other datasets, even larger ones. While competing datasets may excel in specific tasks (e.g., OpenMathInstruct2 on math) but regress in others (mixed green and red cells), CODEI/O shows consistent improvements (mainly green patterns). Despite using only code-centric data, it enhances all other tasks beyond code reasoning as well, suggesting its generalizable capabilities. We also observe that training on raw code files (PythonEdu) results in only minor, and occasionally even negative, improvements compared to the

single-stage baseline, significantly underperforming when compared to CODEI/O, suggesting that learning from such less-structured data is suboptimal. This further highlights that performance gains are driven not merely by data size but by thoughtfully designed training tasks that encompass diverse, structured reasoning patterns in generalized CoTs.

Additionally, CODEI/O++ systematically outperforms CODEI/O, boosting average scores without trade-offs on individual tasks. This highlights how execution-feedback-based multi-turn revision improves data quality and enhances reasoning across domains. Most importantly, both CODEI/O and CODEI/O++ demonstrate performance improvements across models of various sizes and architectures on most benchmarks, although we also observed nearly unchanged or even decreased performance on a small number of tasks. The further validates that our training approach, predicting code inputs and outputs, enables models to excel in diverse reasoning tasks without sacrificing specialized benchmark performance.

### 4. Analysis

To examine the influence of different critical aspects of our approach, we carry out multiple analysis experiments. Unless explicitly stated otherwise, all experiments are performed using Qwen 2.5 Coder 7B for simplicity, and the results reported are those obtained after the second-stage general instruction tuning.

### 4.1. Ablation Studies

We first perform two key ablation studies on our data construction process, with results presented in Table 2:

**Input/Output Prediction** We examine input and output prediction by training on each separately. The scores are generally similar, but input prediction excels on KorBench while slightly hurting GPQA, and output prediction shows greater benefits on symbolic reasoning tasks like BBH. CRUXEval-

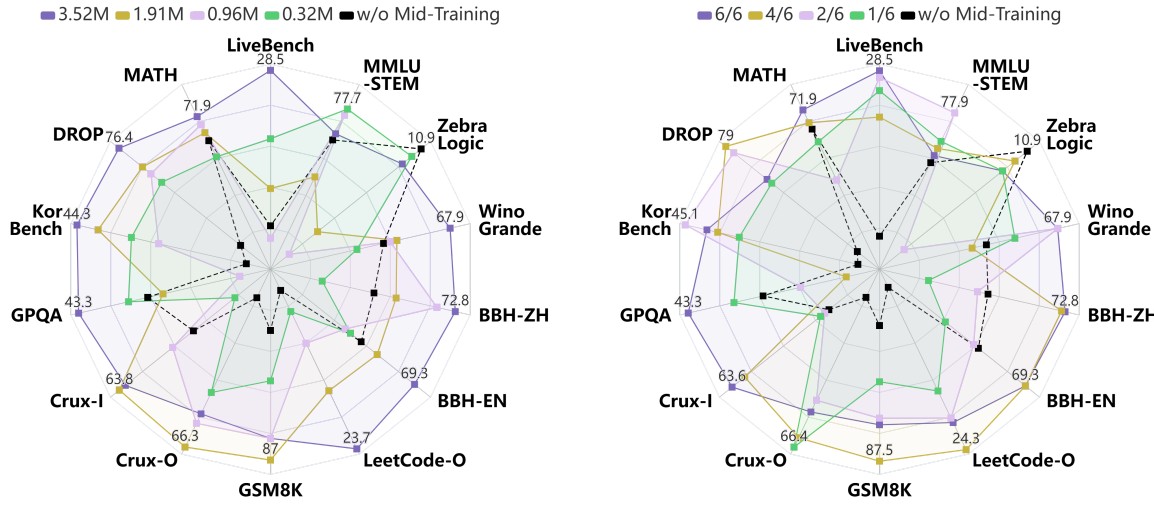

(a) Size of randomly sampled subset.

(b) Ratio of testcases per sample compared to the full set.

Figure 4: The scaling effect of CODEI/O in the first stage training.

I and -O also favor input and output prediction, respectively.

**Rejection Sampling** We explore filtering incorrect responses using rejection sampling, which removes 50% of the training data. However, this results in a general performance drop, suggesting a loss of data diversity. We also experiment with replacing all incorrect responses with ground-truth answers through code execution (without CoT). We see improvements on benchmarks like LeetCode-O and CRUXEval-O designed to measure output prediction accuracy, but it lowers scores elsewhere, reducing the average performance. When comparing these two with training on a ∼ 50% subset of CODEI/O where the number of samples are comparable, they still have no advantages. Therefore, to maintain performance balance, we retain all incorrect responses in the main experiments without any modification.

### 4.2. Effect of Different Synthesis Model

Some of our baselines such as WebInstruct synthesize responses with Qwen-72B (Bai et al., 2023) and Mixtral 22Bx8 (Jiang et al., 2024a), while CODEI/O uses DeepSeek-V2.5. To ablate the effect of different synthesis models, we regenerate responses for the 3.5M WebInstruct (as it covers massive domains) subset using DeepSeek-V2.5, creating an updated dataset called WebInstruct-DS25. As shown in Figure 3, while WebInstruct-DS25 outperforms the vanilla dataset on Qwen 2.5 Coder 7B and LLaMA 3.1 8B, it still falls short of CODEI/O. This highlights the value of diverse reasoning patterns in code and the importance of task selection in training. Overall, this comparison shows that predicting code inputs and outputs improves reasoning beyond mere knowledge distillation from an advanced model.

### 4.3. Scaling Effect of CODEI/O

We evaluate how CODEI/O scales with varying amounts of training data. By randomly sampling training instances, Figure 4a reveals a clear trend: increasing the number of training samples generally leads to improved performance across benchmarks. Specifically, using the smallest amount of data exhibits relatively weak performance on most benchmarks, as the model lacks sufficient training to generalize effectively. In contrast, when trained on the full dataset, CODEI/O achieves the most comprehensive and robust performance. Intermediate amounts of data yield results that fall between these two extremes, demonstrating a gradual improvement in performance as more training samples are introduced. This highlights CODEI/O's scalability and effectiveness in enhancing reasoning capabilities.

We also scale the data on the dimension of input-output pairs by fixing and using all unique raw code samples but changing the number of input-output prediction instances for each sample. Figure 4b shows the ratio of used I/O pairs compared to the full set. While the scaling effect is less pronounced than with training samples, we still observe clear benefits, particularly when increasing from 1/6 to 6/6. This suggests some reasoning patterns require multiple test cases to fully capture and learn their complex logic flow.

### 4.4. Different Data Format

We investigate how to best arrange the query, reference code, and CoT in training samples. As shown in Table 3, placing the query and reference code in the prompt and the CoT in the response achieves the highest average score and most balanced performance across benchmarks. Other formats

Table 3: The effect of different data formats. We make **bold** the highest and underline the lowest scores in each column.

| Data Format | | Wino | | GSM | | | MMLU | LC | CRUX | | BBH | | Zebra | Kor | Live | |
| Prompt | Response | Grande | DROP | 8K | MATH | GPQA | -STEM | -O | -I | -O | -EN | -ZH | Logic | Bench | Bench | AVG |
|---|---|---|---|---|---|---|---|---|---|---|---|---|---|---|---|---|
| Q+Code | CoT | **67.9** | 76.4 | 86.4 | **71.9** | **43.3** | 77.3 | 23.7 | 63.6 | 64.9 | 69.3 | 72.8 | 10.7 | 44.3 | **28.5** | **57.2** |
| Q | CoT | 67.2 | **76.8** | 87.2 | 70.4 | 37.5 | 77.3 | **25.2** | 62.6 | 65.3 | 69.2 | 71.1 | **11.5** | 44.9 | **28.5** | 56.8 |
| Code | CoT | **67.9** | 76.4 | 87.0 | 70.8 | 39.5 | 76.5 | 25.0 | **64.1** | 65.8 | 68.8 | 71.3 | 10.6 | **45.2** | **28.5** | 57.0 |
| Q | Code+CoT | 65.9 | 76.1 | **87.5** | 71.7 | 42.2 | 76.9 | 22.9 | 63.9 | **66.1** | **69.6** | **72.9** | 10.9 | 41.4 | **28.5** | 56.9 |
| Q | Code | 66.9 | 73.1 | 84.8 | 71.6 | 40.0 | **77.4** | 20.8 | 59.5 | 62.4 | 67.2 | 68.3 | 10.1 | 40.3 | 26.3 | 54.9 |

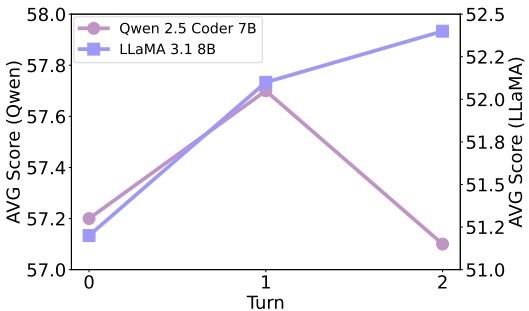

Figure 5: Average benchmark scores from training on data from different turns of revision.

show slightly lower but comparable performance, with the worst results occurring when the query is in the prompt and the reference code in the response, resembling a standard code generation task but with much fewer training samples. This highlights the importance of CoT and the scaling of test cases for learning transferable reasoning ability.

### 4.5. Multi-turn Revision

Based on CODEI/O (no revision) and CODEI/O++ (single-turn revision), we extended revisions to a second turn to evaluate further improvements by regenerating predictions for instances still incorrect after the first revision. We visualize the distribution of response types in each turn in Figure 7 in Appendix D. It shows that most correct responses are predicted in the initial turn, with about 10% of incorrect responses corrected in the first-turn revision. However, the second turn yields significantly fewer corrections, we find by checking the cases that the model often repeats the same incorrect CoT without adding new useful information. After incorporating multi-turn revisions, we observe consistent improvement from turn 0 to turn 1 but minimal gains from turn 1 to turn 2 in Figure 5 – showing slight improvement for LLaMA 3.1 8B but regression for Qwen 2.5 Coder 7B. Hence, we stop at single-turn revision, i.e., CODEI/O++, in our main experiments.

### 4.6. The Necessity of Two Stage Training

Lastly, we highlight the necessity of a separate training stage with CODEI/O data by testing both single-stage mixed train-

Table 4: Average benchmark score under different training strategy. IT stands for our instruction-tuning data.

| First | Second | Model | |
| Stage | Stage | Qwen | LLaMA |
|---|---|---|---|
| - | IT | 54.8 | 49.3 |
| - | CODEI/O(10%)+IT | 56.6 | 50.5 |
| CODEI/O+IT | - | 55.9 | 49.7 |
| CODEI/O | IT | 57.2 | 51.2 |
| CODEI/O+IT | IT | 56.8 | 51.5 |
| CODEI/O | CODEI/O(10%)+IT | 57.0 | 52.7 |

ing and two-stage training with different data mixtures. As shown in Table 4, all two-stage variants outperform single-stage training. Meanwhile, the effect of mixing data during two-stage training varies across models. For Qwen 2.5 Coder 7B, the best result is keeping CODEI/O and instruction-tuning data fully separate, while LLaMA 3.1 8B performs better with mixed data, either in the first stage or in the second stage. To simplify our methodology, we use fully separated data in our main experiments, leaving optimal data-mixing strategies for future work.

### 4.7. Discussion on Data Leakage

Considering the diversity of CODEI/O, it is unavoidable to see pattern overlap in both the training data and the test benchmarks. Thus, distinguishing whether the performance gain is from learning similar reasoning patterns or from seeing identical questions in training becomes important. To do so, we conduct a strict 13-gram-based leakage detection on CODEI/O data following Singh et al. (2024). Specifically, if any normed (punctuation, numbers, and blank spaces removed) 13-gram in a test question is found in the whole training data, it will be tagged as potentially leaked. The results are shown in Table 5, and we can see that for most of the benchmarks, there is no risk of leakage, or the leakage ratio is far less than the performance gain.

Upon manual inspection of the two benchmarks – LeetCode-O and KorBench – with high potential leakage ratios, we find that:

1) KorBench overlaps only contain general descriptions like Sudoku rules or common letter sequences ("A B C D...") rather than specific questions – our training tasks and the

Table 5: Potential data leakage ratio over different benchmarks measured by 13-gram overlap.

| Benchmark | Ratio (%) | Benchmark | Ratio (%) |
|-----------|-----------|-----------|-----------|
| LeetCode-O | 21.5 | MMLU | 0.1 |
| KorBench | 5.1 | CRUXEval | 0.1 |
| MATH | 0.1 | Others | 0.0 |

Table 6: Sample-wise accuracy gains of CODEI/O compared to the baseline on both the full set and the non-leaked set.

| Model | LeetCode-O | | KorBench | |
|-------|------|---------|------|---------|
| | full | unleaked | full | unleaked |
| Qwen | 3.8 | 3.9 | 5.8 | 6.1 |
| LLaMA | 9.4 | 9.4 | 1.0 | 0.9 |
| DSLite | 5.3 | 5.7 | -1.2 | -1.3 |
| Gemma | 3.7 | 3.9 | 1.4 | 1.3 |

benchmark tasks are completely different.

2) LeetCode-O overlaps stem from sibling problems sharing common descriptions (e.g., Two Sum I & II), despite our having removed all original problems from our training data.

To further evaluate whether the gains on these two benchmarks are due to data leakage, we calculated the sample-wise accuracy gains of CODEI/O compared to the baseline on both the full set and the non-leaked set for various models. The results are shown in Table 6. The similar gains observed on both full and non-leaked subsets across all models confirm that our improvements are not attributable to data leakage.

## 5. Related Work

**Learning about Code Execution** The topic of learning code execution has existed long before the era of LLMs (Zaremba & Sutskever, 2014; Graves et al., 2014). However, most related works focus solely on the output prediction task itself when learning from code execution (Nye et al., 2021; Liu et al., 2023; Ding et al., 2024c). Other works seek to utilize code execution, either through the final feedback (Ding et al., 2024a; Wang et al., 2024) or the intermediate trace (Ding et al., 2024b; Ni et al., 2024), to improve code generation abilities. There are also specific benchmarks designed to evaluate a model's ability to predict execution results, such as CRUXEval (Gu et al., 2024) and LiveCodeBench-Exec (Jain et al., 2024). Similar to our method, Jiang et al. (2024b) also propose to learn from code execution to enhance general reasoning ability, but the involved functions are mainly LeetCode ones. Unlike the above works, which set a narrow scope within code-related tasks, we are the first

to train LLMs on large-scale, diverse code input-output predictions and demonstrate its efficacy in improving general reasoning ability beyond code.

**Inference Time Scaling** A very recent approach to enhance reasoning is inference-time scaling, such as OpenAI's o1 (Jaech et al., 2024) or DeepSeek's R1 (DeepSeek-AI et al., 2025), which typically encourages models to generate ultra-long reasoning process to solve problems through large-scale reinforcement learning. Such methods are pushing models to new limits on massive challenge tasks, while also significantly altering the output patterns of models. We believe that CODEI/O is orthogonal to these methods, and we hope it can provide a better basis to further incentivize the reasoning abilities of LLMs.

## 6. Conclusion

In conclusion, we introduced CODEI/O, an approach to improve the reasoning abilities of LLMs by training them to predict code inputs and outputs in pure natural language CoTs. This approach leverages the structured and scalable nature of code to learn diverse reasoning patterns, including symbolic, logical, mathematical, and commonsense reasoning. Extensive experiments show that CODEI/O as well as the enhanced CODEI/O++ consistently outperforms existing baselines, delivering balanced improvements across benchmarks without sacrificing performance in any domain, underscoring its robustness and versatility.

## Acknowledgments

We thank the anonymous reviewers for their valuable comments. This work is largely inspired by a previously unpublished project, and we hereby thank its core participants: Zhoujun Cheng, Fan Zhou, Yu Gu, and Qian Liu. We also thank Wei Liu and Yiheng Xu for their valuable feedback and suggestions on paper writing.

## Impact Statement

This paper presents work whose goal is to advance the field of Machine Learning. There are many potential societal consequences of our work, none which we feel must be specifically highlighted here.

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

# A. Details of Checking Execution Complexity

During the execution of these codes, we set a runtime limit of 5 seconds for each sample. Additionally, we impose constraints on the complexity of the input and output objects to ensure they remain predictable and within the generation capability of general LLMs: total size of objects must be less than 1024 bytes, length of lists and dictionaries should be less than 20, and strings should be no longer than 100 characters. For objects other than these simple types, we enforce a size limit of 128 bytes. These checks are conducted recursively to ensure that all sub-objects within the full input/output object comply with these constraints. The exact code for these checks is shown below:

```python
from pympler import asizeof

def strict_check_size(obj):
    if asizeof.asizeof(obj) >= 1024:
        return False
    if isinstance(obj, dict):
        if len(obj) >= 20:
            return False
        for k, v in obj.items():
            if not strict_check_size(k) or not strict_check_size(v):
                return False
    elif isinstance(obj, (list, tuple, set)):
        if len(obj) >= 20:
            return False
        for item in obj:
            if not strict_check_size(item):
                return False
    elif isinstance(obj, str):
        if len(obj) >= 100:
            return False
    else:
        if asizeof.asizeof(obj) >= 128:
            return False
    return True
```

# B. Details of Selected Benchmarks

We introduce the details of all the benchmarks we use in this work. The sizes of their test sets are shown in Table 7. Some parts of these descriptions largely refer to Yue et al. (2024). The following are the established ones:

**WinoGrande** (Sakaguchi et al., 2020): WinoGrande is a benchmark for commonsense reasoning with expert-crafted pronoun resolution problems.

**DROP** (Dua et al., 2019): DROP is a benchmark for numerical reasoning in reading comprehension. It demands resolving references and performing operations like addition, counting, or sorting. We report the F1 score as the metric.

**GSM8K** (Cobbe et al., 2021): GSM8K contains diverse grade-school math problems, intended to test basic arithmetic and reasoning abilities in an educational context.

**MATH** (Hendrycks et al., 2021b): MATH comprises intricate competition-level problems across 5 levels to evaluate the models' ability to perform complex mathematical reasoning.

**GPQA** (Rein et al., 2024): GPQA provides "Google-proof" questions in biology, physics, and chemistry, designed to test deep domain expertise and reasoning under challenging conditions. We use its complete set.

**MMLU-STEM** (Hendrycks et al., 2021a): MMLU spans 57 subjects across multiple disciplines. MMLU evaluates the breadth and depth of a model's knowledge in a manner akin to academic and professional testing environments. We select the STEM subset of MMLU.

**CRUXEval** (Gu et al., 2024): CRUXEval is designed to test a model's ability to predict the inputs or outputs given an anonymized Python function.

**BBH** (Suzgun et al., 2023): BBH consists of 23 tasks previously found challenging for language models from BIG-Bench (Srivastava et al., 2022).

Table 7: The number of test samples in the test set of each benchmark.

| Wino Grande | DROP | GSM 8K | MATH | GPQA | MMLU -STEM | LC -O | CRUX -I | CRUX -O | BBH -EN | BBH -ZH | Zebra Logic | Kor Bench | Live Bench |
|---|---|---|---|---|---|---|---|---|---|---|---|---|---|
| 1,267 | 9,536 | 1,319 | 5,000 | 448 | 3,153 | 900 | 800 | 800 | 6,511 | 2,250 | 1,000 | 1,250 | 672 |

```
You have n coins and you want to build a staircase with these coins. The staircase consists of k
rows where the ith row has exactly i coins. The last row of the staircase may be incomplete.

Given the integer n, return the number of complete rows of the staircase you will build.

n = 5

What is the Output?  Solve the problem without writing any code, and present your final
answer at the end in the following json format: {"output": <my output>}.
```

Figure 6: An example in the constructed Leetcode-O benchmark.

**ZebraLogic** (Lin et al., 2025): ZebraLogic is a benchmark using Logic Grid Puzzles (Zebra Puzzles) to test LLMs' logical reasoning. It involves deducing unique feature assignments for N houses based on clues, evaluating Constraint Satisfaction Problem (CSP) solving skills, similar to human reasoning tests like the LSAT.

**KorBench** (Ma et al., 2024): KorBench is designed to evaluate models' intrinsic reasoning and planning abilities by minimizing interference from pretrained knowledge. It introduces new rules that are independent of prior knowledge, allowing for a more accurate assessment of how models adapt to novel rule-driven tasks. KorBench consists of five task categories: Operation, Logic, Cipher, Puzzle, and Counterfactual, each containing 25 manually defined rules.

**LiveBench** (White et al., 2024): LiveBench is a benchmark for LLMs designed to prevent test set contamination and ensure objective evaluation. It releases new questions monthly, sourced from recent datasets, arXiv papers, news, and IMDb synopses. Each question has verifiable, objective answers, enabling accurate, automatic scoring without LLM judges. It includes 18 diverse tasks across 6 categories, including reasoning, coding, mathematics, data analysis, language, and instruction following.

We also have two extra ones created by ourselves:

**BBH-ZH**: We translate 9 tasks from BBH, i.e., *boolean expressions, date understanding, logical deduction five objects, navigate, object counting, reasoning about colored objects, temporal sequences, tracking shuffled objects five objects, web of lies*, to build a Chinese version.

**LeetCode-O**: We build a benchmark for testing code output prediction based on LeetCode problems. Different from CRUXEval, which only provides the Python function but not the textual query, LeetCode-O instead only provides the textual query but not the function, making it more challenging as the model needs to come up with the implicit solution by itself. In constructing this benchmark, we deliberately skip all LeetCode problems that have been covered by our training dataset and keep a balanced distribution of 300/300/300 across easy/medium/hard problems. For each problem, we also have both the original English version and a Chinese-translated version. We report the problem-level accuracy, which gets 1 point if and only if the model correctly predicts the outputs for all inputs under both languages; otherwise, it gets 0 points. The input-output pairs are collected solely from the original problem descriptions; thus, each problem usually has 2 or 3 test cases. We provide an example for this benchmark in Figure 6.

## C. Details of Processing Different Data Sources

### C.1. Source Distribution

**CodeMix** CodeMix is a large collection of raw Python code files retrieved and curated from an in-house code pre-training corpus. To ensure the quality and relevance of the dataset, we filter out files that are either overly simplistic or excessively complex. This filtering process is based on the success rate of the DeepSeek-Coder-V2-Lite-Inst model in doing a function completion task derived from each file. Files with a success rate between 10% and 90% are retained, resulting in a collection of approximately 427K code files.

Table 8: Ablation study on data sources, we remove all training samples originate from one major data source.

| | # (M) | Wino Grande | DROP | GSM 8K | MATH | GPQA | MMLU -STEM | LC -O | CRUX -I | CRUX -O | BBH -EN | BBH -ZH | Zebra Logic | Kor Bench | Live Bench | AVG |
|---|---|---|---|---|---|---|---|---|---|---|---|---|---|---|---|---|
| CODEI/O | 3.52 | 67.9 | 76.4 | 86.4 | 71.9 | 43.3 | 77.3 | 23.7 | 63.6 | 64.9 | 69.3 | 72.8 | 10.7 | 44.3 | 28.5 | 57.2 |
| ~ 50% subset | 1.59 | 67.5 | 74.7 | 86.7 | 71.6 | 42.9 | 77.3 | 23.0 | 62.8 | 65.9 | 69.1 | 70.8 | 10.5 | 42.1 | 28.9 | 56.7 |
| w/o CodeMix | 1.84 | 65.8 | 76.6 | 87.3 | 70.9 | 42.6 | 77.0 | 21.8 | 62.0 | 65.0 | 68.5 | 69.5 | 10.7 | 43.8 | 26.8 | 56.3 |
| w/o PyEdu-R | 1.89 | 66.8 | 75.4 | 86.0 | 71.4 | 40.6 | 77.0 | 24.1 | 61.8 | 64.8 | 69.8 | 72.3 | 11.0 | 46.3 | 30.1 | 57.0 |

**PyEdu-R** Python-Edu (Ben Allal et al., 2024) is a large dataset containing about 7.7M high-quality Python code files sourced from the Stack-V2 dataset (Lozhkov et al., 2024). These files are annotated with an additional scoring model to evaluate their educational quality. Since our analysis indicates that a significant portion of the CodeMix dataset focuses on algorithms, we intentionally exclude similar content from Python-Edu. To achieve this, we classify all code files into several categories, such as algorithms, logic puzzles, math-related tasks, scientific computation, system modeling, other complex reasoning, and non-reasoning codes, using both DeepSeek-Coder-V2-Lite-Inst and DeepSeek-V2.5. We remove files classified as algorithms and non-reasoning. The resulting subset is referred to as PyEdu-R(easoning) and contains approximately 369K code files.

**Other Sources** In addition to the two major datasets described above, we also collect high-quality code files from a variety of other reputable sources. These include comprehensive algorithm repositories[3], challenging mathematical problem collections[4], and well-known online coding platforms[5]. After consolidating all of these sources, we obtain a total of approximately 14.5K code files.

## C.2. Input-Output Pairs for Each Source

**CodeMix** For each sample in this subset, we select at most 3 pairs of input-output examples, resulting in 3 input prediction instances and 3 output prediction instances per sample. After filtering, we obtain 300K samples, with an average of 2.78 input and 2.80 output prediction instances per sample, totaling 1,674,345 instances.

**Pyedu-R** For each sample in this subset, we select at most 6 pairs of input-output examples, resulting in 6 input prediction instances and 6 output prediction instances per sample. After filtering, we obtain 141K samples, with an average of 5.77 input and 5.79 output prediction instances per sample, totaling 1,630,716 instances.

**Other Sources** For each sample in this subset, we select at most 10 pairs of input-output examples, resulting in 10 input prediction instances and 10 output prediction instances per sample. After filtering, we obtain 13.9K samples, with an average of 7.70 input and 7.87 output prediction instances per sample, totaling 216,159 instances.

## C.3. The Effect of Using Different Sources

We analyze the contributions of our two main data sources, CodeMix and PyEdu-R, by excluding the training samples from each. The results, presented in Table 8, indicate that removing PyEdu-R reduces performance on mathematical and scientific benchmarks (e.g., DROP, GSM8K, GPQA), consistent with its construction process. In contrast, removing CodeMix has a greater negative impact on symbolic or logical tasks, reflecting its focus on algorithmic content.

Nevertheless, combining both data sources yields the best overall performance. When comparing them to a similarly sized subset of CODEI/O, we observe that removing CodeMix results in a performance decline, while removing PyEdu-R has a smaller effect. Upon further inspection of the samples from PyEdu-R, we find that many focus on complex calculations involving nontrivial floating-point numbers, but place less emphasis on high-level reasoning or problem-solving flows. This characteristic makes PyEdu-R challenging for models to learn from effectively. Future work could explore refining or cleaning PyEdu-R to enhance its learnability and utility.

---

[3] https://github.com/TheAlgorithms/Python
[4] https://projecteuler.net
[5] https://github.com/doocs/leetcode, https://www.codewars.com, https://edabit.com, https://codeforces.com, https://atcoder.jp, https://www.codechef.com

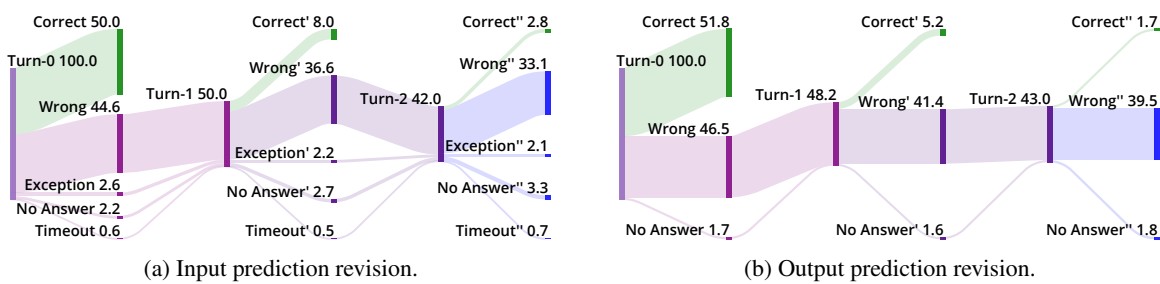

(a) Input prediction revision.  (b) Output prediction revision.

Figure 7: In multi-turn revision, we track the percentage (%) of each response type across the entire dataset after each revision turn.

Table 9: Performance when switching the second stage instruction-tuning data to Tulu-3 (Lambert et al., 2024).

| Stage 1 | Stage 2 | Wino Grande | DROP | GSM 8K | MATH | GPQA | MMLU -STEM | LC -O | CRUX -I | -O | BBH -EN | -ZH | Zebra Logic | Kor Bench | Live Bench | AVG |
|---|---|---|---|---|---|---|---|---|---|---|---|---|---|---|---|---|
| - | Tulu3 | 66.9 | 59.9 | 81.0 | 38.3 | 27.2 | 69.3 | 16.6 | 53.9 | 59.9 | 63.9 | 69.8 | 8.9 | 35.2 | 22.5 | 48.1 |
| CODEI/O | Tulu3 | 68.1 | 59.4 | 79.3 | 39.8 | 30.4 | 71.3 | 17.6 | 57.1 | 64.3 | 65.8 | 72.5 | 8.6 | 43.4 | 22.4 | 50.0 |
| CODEI/O++ | Tulu3 | 67.6 | 54.9 | 80.5 | 39.9 | 27.5 | 71.9 | 20.6 | 57.3 | 63.3 | 66.6 | 70.0 | 9.3 | 41.8 | 24.3 | 49.7 |

## D. Detailed Statistics in Multi-turn Revision

We report the detailed distribution of response types after each revision turn in Figure 7, including both input and output predictions. In general, we observe that most correct predictions are already made in the initial turn. During the first revision turn, approximately 16% and 10% of incorrect predictions are revised for input and output predictions, respectively. However, in the second revision turn, significantly fewer predictions are revised, indicating a rapidly diminishing benefit from further revisions. Therefore, to save computing resources—both in prompting DeepSeek-V2.5 and executing the code for verification—we limit the revision process to a single turn in our main experiments.

## E. Training Hyper-parameters

During the first stage, we train for 1 epoch using a constant learning rate, which is set to 1e-5 for the three smaller models and 4e-6 for Gemma 2 27B. The batch size is 1024. In the second stage, we train for 700 steps with the batch size of 1024 as well, corresponding to about 3 epochs of the entire instruction-tuning dataset. The learning rate is set to 3e-5 for the three smaller models and 1e-5 for Gemma 2 27B, using a cosine scheduler decaying to 1e-6 and 3e-7, respectively. In both training stages, no warmup period is applied and the maximum sequence length is set to 4096.

## F. The Effect of Using Other Instruction-Tuning Data

Besides our in-house instruction-tuning data, we also conduct experiments using another strong public instruction-tuning dataset, Tulu-3 (Lambert et al., 2024). We present the benchmark performance when training on Qwen 2.5 Coder 7B in Table 9. After switching to this new dataset for the second stage of training, we still observe a significant improvement over the single-stage baseline, indicating the robustness of CODEI/O. However, contrary to our main experiments, CODEI/O++ performs slightly worse than CODEI/O. A potential reason for this could be that Tulu-3 does not cover as many instruction types as our in-house dataset, which may limit the model's ability to fully leverage its reasoning capabilities after training.

## G. Examples Mentioned in the Main Text

In this section, we present the examples mentioned in the main text. Table 10 illustrates an example of how we transform a raw code file into our desired unified format. Figure 8 provides an example of the exact prompt used in both response collection and training. Additionally, Table 11 demonstrates a complete training sample in CODEI/O++. In this sample, the initial response is incorrect, but after incorporating feedback and regenerating the response, DeepSeek-V2.5 successfully revise it to a correct prediction at the second turn.

Table 10: A complete example showing how we transform a raw code file into our designed unified format.

| Raw Code File | Cleaned Reference Code (with Main Entrypoint Function) |
|---|---|

```
#get the vertical acceleration data
acceleration [. . . . . . . . ]
# pass acceleration data to low pass filter
time[. . . . . . . . . . . ]

#code to find speed at each point
initial_speed = current_speed
delta_t = t_current  - t_prev
curr_acc = acc[i]
current_speed = initial_speed + (curr_acc) * delta_t

#code to find dispacement

initial_dis = current_disp
delta_t = t_current  - t_prev
curr_speed = ("call above algorithm")
current_disp = initial_dis + (initial_speed + current_speed)/2
↪ * delta_t ;

#code to find horizontal displacement

#use the above same code and find horizontal displacement
#pass both the displacement data through low pass filter
#map horizontal and vertical displacement to give road profile
```

```python
# import necessary packages
import numpy as np

# main function
def main_solution(acceleration, time, initial_speed,
↪ initial_displacement):
    # Convert inputs to numpy arrays if they are not already
    acceleration = np.array(acceleration)
    time = np.array(time)

    # Initialize variables
    current_speed = initial_speed
    current_disp = initial_displacement

    # Calculate speed and displacement
    speeds = []
    displacements = []

    for i in range(1, len(time)):
        delta_t = time[i] - time[i-1]
        curr_acc = acceleration[i]
        current_speed = current_speed + curr_acc * delta_t
        speeds.append(current_speed)

        current_disp = current_disp + (initial_speed +
        ↪ current_speed) / 2 * delta_t
        displacements.append(current_disp)

        initial_speed = current_speed

    # Convert outputs to JSON serializable format
    speeds = [float(speed) for speed in speeds]
    displacements = [float(disp) for disp in displacements]

    return {"speeds": speeds, "displacements": displacements}
```

### Query

Given a set of vertical acceleration data and corresponding time points, how can we determine the speed and displacement of a vehicle at each time point, starting from an initial speed and displacement?

| Input/Output Description | Input Generator |
|---|---|

```
Input:
    acceleration (list of float): List of vertical
    ↪ acceleration values at each time point.
    time (list of float): List of time points corresponding to
    ↪ the acceleration values.
    initial_speed (float): Initial speed at the first time
    ↪ point.
    initial_displacement (float): Initial displacement at the
    ↪ first time point.

Output:
    return (dict): A dictionary containing two keys:
    - speeds (list of float): List of calculated speeds at
    ↪ each time point.
    - displacements (list of float): List of calculated
    ↪ displacements at each time point.
```

```python
def input_generator():
    # Generate random acceleration data
    acceleration = [np.random.uniform(-10, 10) for _ in
    ↪ range(10)]

    # Generate corresponding time data
    time = [0.1 * i for i in range(10)]

    # Generate initial speed and displacement
    initial_speed = np.random.uniform(0, 10)
    initial_displacement = np.random.uniform(0, 10)

    return {
        "acceleration": acceleration,
        "time": time,
        "initial_speed": initial_speed,
        "initial_displacement": initial_displacement
    }
```

```
You are given a question that requires some input and output variables as follows:

Given two jugs with capacities of 'x' and 'y' liters, is it possible to measure exactly
'z' liters of water using these two jugs? What is the result of this measurement attempt?

The input and output requirements are as follows:

Input:
  'x' (int): The capacity of the first jug in liters.
  'y' (int): The capacity of the second jug in liters.
  'z' (int): The desired amount of water to measure in liters.

Output:
'return' (bool): True if it is possible to measure exactly z liters using the two jugs, False
otherwise.
```

Given the following output:                                                   Input Prediction

True

Can you predict a feasible input without writing any code?  Please reason and put your
final answer in the following json format: "input": <your input>, where <your input> should be
a dictionary, even if the there is only one input variable, with keys strictly match the input
variables' names as specified.

Given the following input:                                                    Output Prediction

{"x":5, "y": 6, "z": 7}

Can you predict the output without writing any code?  Please reason and put your final
answer in the following json format: "output": <your output>, where <your output> should strictly
match the the output requirement as specified.

Tip:  Here is a reference code snippet for this question.  You can refer to this code to
guide your reasoning but not copy spans of code directly.

```python
# import necessary packages
import random

# main function
def main_solution(x, y, z):
    """
    Determines if it is possible to measure exactly z liters using two jugs with capacities x and y
    liters.

    :param x: int, capacity of the first jug in liters
    :param y: int, capacity of the second jug in liters
    :param z: int, the desired amount of water to measure in liters
    :return: bool, True if it is possible to measure exactly z liters, False otherwise
    """
    if z == 0:
        return True
    if x + y < z:
        return False
    if x > y:
        x, y = y, x
    if x == 0:
        return y == z
    while y % x != 0:
        y, x = x, y % x
    return z % x == 0
```

Figure 8: An example prompt we used for collecting responses and training. Most parts of the prompt for input and output prediction remain the same, except for the two gray background blocks, which is used as needed.

Table 11: A complete example for input prediction in CODEI/O++ training data. The reference code is omitted in the **prompt**, and the given input is highlighted. For the **concatenated response**, we highlight the template, the Turn 1 feedback, and the Turn 2 feedback, and all the other contents are generated by DeepSeek-V2.5 in two turns.

---

**Prompt**

You are given a question that requires some input and output variables as follows:

Given a list of integers numbers and an integer target, determine the length of the shortest contiguous subarray whose sum is at least target. If no such subarray exists, return 0. How can you find this shortest subarray length efficiently?

The input and output requirements are as follows:

Input:
  - target (int): The target sum that the subarray should at least reach.
  - numbers (list of int): A list of integers representing the array from which the subarray is to be found.

Output:
  - return (int): The length of the shortest contiguous subarray whose sum is at least the target. Returns 0 if no such subarray exists.

Given the following output: 4

Can you predict a feasible input without writing any code? Please reason and put your final answer in the following json format: {"input": <your input>}, where <your input> should be a dictionary with keys strictly match the input variables' names as specified.

---

**Concatenated Response (Turn 1 generation + Turn 1 feedback + Turn 2 generation + Turn 2 feedback)**

To predict a feasible input that results in the output 4, we need to consider the following:

1. **Target Sum**: The target sum is the minimum sum that a subarray should achieve.
2. **Numbers List**: The list of integers from which we need to find the shortest subarray.
3. **Output**: The length of the shortest subarray whose sum is at least the target.

Given the output is 4, this means the shortest subarray that meets or exceeds the target sum has a length of 4.

### Reasoning:
- **Target Sum**: Let's assume the target sum is 10.
- **Numbers List**: We need to construct a list where the shortest subarray that sums to at least 10 has a length of 4.

### Example Construction:
- If we have a list like [1, 2, 3, 4, 5], the subarray [2, 3, 4, 5] sums to 14, which is greater than 10, and its length is 4.

Thus, a feasible input that results in the output 4 could be:

```json
{"input": {"target": 10, "numbers": [1, 2, 3, 4, 5]}}
```

This input satisfies the condition that the shortest subarray with a sum of at least 10 has a length of 4.

Let me check if I did it correctly ...... Oops! Something went wrong and I find this

[Mismatch] Your input is not feasible! Given the output 4, your predicted input is {"target": 10, "numbers": [1, 2, 3, 4, 5]}, which actually gets a wrong output as 3

Well ...... I apologize for the oversight. Let's re-evaluate the input to ensure it matches the output 4.

We need to find a list of integers where the shortest contiguous subarray whose sum is at least target has a length of exactly 4.

### Revised Reasoning:
- **Target Sum**: Let's keep the target sum as 10.
- **Numbers List**: We need a list where a subarray of length 4 sums to 10 or more.

### Example Construction:
- Consider the list [1, 2, 2, 2, 2, 2]:
  - The subarray [2, 2, 2, 2] sums to 8, which is less than 10.
  - However, if we add another 2, making it [1, 2, 2, 2, 2, 2, 2], the subarray [2, 2, 2, 2] still sums to 8.

- Let's try a different approach with larger numbers to ensure the subarray of length 4 sums to at least 10:
  - Consider the list [1, 3, 2, 2, 5, 1]:
    - The subarray [1, 3, 2, 2] sums to 8, which is less than 10.
    - The subarray [3, 2, 2, 5] sums to 12, which is more than 10, and its length is 4.

Thus, a feasible input that results in the output 4 could be:

```json
{"input": {"target": 10, "numbers": [1, 3, 2, 2, 5, 1]}}
```

This input satisfies the condition that the shortest subarray with a sum of at least 10 has a length of 4.

Let me check if I did it correctly ...... Yes, that's correct! I made it!

---

