# OpenReview forum: "CodeIO: Condensing Reasoning Patterns via Code Input-Output Prediction"
_ICML.cc/2025/Conference — ICML 2025 oral_

### Official Review · Reviewer_z6HH · 2025-03-08

**Overall Recommendation:** 3

**Summary:**

The paper reports on work on generating training data for reasoning tasks from code. The method generates training examples from code by using a LLM to generate the query, input-output pairs with their reasoning chains, and input predictions from the output together with their reasoning chains. The data is used for fine tuning a LLM before instruction turning. Experiments show that the method obtain improved performance on many reasoning benchmarks for multiple LLMs.

## update after rebuttal
Thanks for the rebuttal. From the rebuttal, it seems that the process is creating more training examples to reinforce other data sources, rather than data that create behaviour change. I still think this is useful work and maintain my score.

**Claims And Evidence:**

The paper claims that the training examples constructed as they describe in the paper provide universal reasoning primitives. Experiments show consistent improvements over many benchmarks, hence support a claim that the method is useful, although "universal" is a strong claim and is not defined in the paper.

**Essential References Not Discussed:**

I am not aware of essential references that are not discussed.

**Experimental Designs Or Analyses:**

Multiple ablation studies are done and they appear appropriate.

**Methods And Evaluation Criteria:**

The experiments are done over a wide range of reasoning benchmarks and LLMs, hence seems appropriate.

**Other Comments Or Suggestions:**

It would be useful for the authors to summarize the list of insights gained from their work in the introduction.

**Other Strengths And Weaknesses:**

The paper shows that code can be used to training examples that are useful for learning to reason. Of interest is the particular types of training examples that are useful. The paper shows that one type of useful example generated using code is example inputs together with the chain of thought to generate the output. This is interesting but somewhat expected. The other reasoning pattern that is useful is less expected -- from an output generate a valid input, with the corresponding chain of thought.

**Questions For Authors:**

What are the helpful reasoning patterns provided by the examples that are generated from code? A qualitative study of validation set (not test set) problems that change from incorrect to correct after inclusion of the new training set may provide useful insights.

**Relation To Broader Scientific Literature:**

There is a lot of work on code generation but to my knowledge, the use of code to generate examples to improve general reasoning is new.

**Theoretical Claims:**

No theoretical claims in the paper.

---

> ### Author Rebuttal · Authors · 2025-04-01
>
> Thank you for the time and effort you spent reviewing our paper, and for recognizing our contributions. Below are our responses:
> # Q1
> > "Universal gains/effectiveness" is a strong claim and is not defined in the paper.
>
> Thanks for this comment, we will change this statement containing “universal” to a more precise one, e.g., “CodeI/O and CodeI/O++ demonstrate performance improvements across models of various sizes and architectures on most benchmarks, although we also observed nearly unchanged or even decreased performance on a small number of tasks.”
> # Q2
> > Summarize the list of insights.
>
> Thanks for the suggestion. We try to summarize some as follows:
>
> 1) Learning reasoning patterns via input/output prediction in natural language offers more general and robust reasoning abilities than raw code pre-training or function generation.
>
> 2) Code's verifiability improves data quality through regenerating incorrect predictions and concatenating multi-turn responses, encouraging model self-reflection.
>
> 3) Learning from code via input/output prediction enhances broader downstream reasoning tasks compared to knowledge-intensive instruction tuning (e.g., WebInstruct, OpenCoder-SFT) or single-domain data (e.g., OpenMathInstruct-2).
>
> We will merge them in the introduction and update our paper accordingly in the next version.
> # Q3
> > Helpful reasoning patterns in CodeI/O and qualitative study.
>
> We provide some examples of generated responses in the CodeI/O dataset with certain reasoning patterns as follows:
>
> *Visual aided reasoning:*
> ```
> …
> We can visualize the tree structure as follows:
>
>         1
>        / \
>       1   1
>      /     \
>     None   1
>     / \     / \
>   None None 1 1
>            / \
>           None 1
> ...
>
> ```
>
> *Verification and self-correction:*
> ```
> …
> 5. Fifth Attempt:
> Let’s try poly=[4,6,10] and x=2.
> Using Horner’s method:
> \[
> ret=4
> ret=4*2+6=14
> red=14*2+10=38
> \]
> This gives us 38, which is close but still not exact.
> 6. Final Attempt:
> Let’s try poly=[4,6,10.7] and x=2.
> Using Horner’s method:
> \[
> ret=4
> ret=4*2+6=14
> red=14*2+10.7=38.7
> \]
> This gives us the exact output 38.7.
> Thus, the feasible input is {“input”: {“poly”: [4,6,10.7], “x”:2}
> ```
>
> *State-space searching and backtracking:*
> ```
> …
> #### DFS Execution:
> - Start from node 0:
>     - Sum at node 0: `1`
>     - Move to node 2:
>     - Sum at node 2: `1 + 1 = 2`
>     - Move to node 1:
>         - Sum at node 1: `2 + 8 = 10`
>         - Move to node 3:
>         - Sum at node 3: `10 + 4 = 14`
>         - Move back to node 1:
>         - Sum at node 1: `10` (backtrack)
>     - Move to node 4:
>         - Sum at node 4: `2 + 4 = 6`
>     - Move back to node 2:
>         - Sum at node 2: `2` (backtrack)
>     - Move back to node 0:
>     - Sum at node 0: `1` (backtrack)
> …
> ```
>
> *Decision tree traversal:*
> ```
> …
> - 197: Check if it is prime.
>     - 197 is not divisible by 2 (it's odd).
>     - Check divisibility by 3, 5, 7, etc., up to the square root of 197 (approximately 14.03).
>     - 197 is not divisible by 3 (197%3!=0).
>     - 197 is not divisible by 5 (197%5!=0).
>     - 197 is not divisible by 7 (197%7!=0).
>     - 197 is not divisible by 11 (197%11!=0).
>     - 197 is not divisible by 13 (197%13!=0).
>     - Since 197 is not divisible by any of these numbers, it is a prime number.
> …
> ```
>
> *Sub-task decomposition*
> ```
> …
> Given the output list [27, 19, 28, 18, 25, 54]
> …
> Let's break down each output value:
>
> 1. For 27:
>     27=4X+Y
>     Possible pairs (X, Y) that satisfy this equation:
>     - X=6, Y=3 (since 4*6+3=27)
>
> 1. For 19:
>    19=4X+Y
>     Possible pairs (X, Y) that satisfy this equation:
>     - X=4, Y=3 (since 4*4+3=19)
> …
> ```
>
> These examples in CodeI/O show that diverse reasoning patterns can be captured in the training set. However, there is no significant behavioral change when comparing models that are only instruction-tuned (baseline) and models with extra first-stage CodeI/O training. For example, most changes from incorrect to correct answers are due to avoiding simple and obvious mistakes, as follows:
>
> *Question:*
> ```
> Jenna starts out with 8 sapphires. She trades 3 sapphires for two rubies. If sapphires are worth $800 and rubies are worth $1200, how much money are all her jewels worth?
> ```
>
> *Baseline wrong response (it does not notice that 3 sapphires have been traded):*
> ```
> …
> Adding the value of her sapphires and rubies, the total value of all her jewels is $6400 (sapphires) + $2400 (rubies) = $8800.
> So the answer is $\boxed{8800}$.
> ```
>
> *CodeI/O correct response:*
> ```
> …
> So, after the trade, Jenna has $6400 - $2400 = $4000 worth of sapphires left.
> …
> Adding the value of her sapphires and rubies together, Jenna's total worth of jewels is $4000 + $2400 = $6400.
> So the answer is $\boxed{6400}$.
> ```
>
> We hypothesize that 2nd stage instruction tuning on shared data leads models to converge to similar states rather than develop distinct response patterns. More in-depth analysis is needed to further understand this subtle behavior change, and we leave it as important future work.

---

### Official Review · Reviewer_rx1K · 2025-03-10

**Overall Recommendation:** 3

**Summary:**

The paper introduces a training paradigm where models are taught to predict input–output pairs from code and accompanying test cases. The key idea is to leverage the structured nature of code to instill reasoning skills while preserving procedural rigor. In practice, the authors transform raw code into executable functions and frame tasks as either predicting execution outputs from a given function and query, or inferring feasible inputs from desired outputs. They also incorporate a multi-turn revision mechanism intended to correct initial errors. Experimental results, including comparisons of query + code versus chain-of-thought (CoT) + code setups, demonstrate promising improvements—though the gains from additional revision turns appear to taper off.

## Update after rebuttal

I confirm that I have read the author response and i appreciate the detailed, easy-to-follow way of rebuttal. I still hold a positive opinion about this paper, thanks!

**Claims And Evidence:**

The paper claims that using structured code formats for input–output prediction can better capture reasoning signals compared to conventional pre-training on raw code. It also asserts that multi-turn revision improves output quality by correcting initial errors. However, while there is experimental evidence supporting some of these claims (e.g., performance gains in certain setups), concerns remain:
* The evidence that diverse reasoning patterns—especially those not naturally aligned with procedural logic—is fully captured remains limited.
* The diminishing gains after the first revision turn suggest that the multi-turn mechanism may not robustly address error propagation.
* Reliance on generated responses (via Deepseek-v2.5) raises questions about the consistency and validity of the training data.

**Essential References Not Discussed:**

I think the papers might want to discuss more regarding the recent advances in multi-turn revision or iterative refinement to address the error propagation in LLM outputs.

**Experimental Designs Or Analyses:**

I have checked the experiment designs and I think it is valid and extensional. One concern is that in Table 3, I’m curious about the performance of using query+code in the prompt, and let the response be both Cot+Code? Since it seems like these two setups have both promising results across different metrics.

**Methods And Evaluation Criteria:**

I think the methods are sound at hand. The proposed method focus on recast code execution as a NL input-output prediction task. Several key elements are (1) transforming code into executable functions and defining dual prediction tasks (output prediction and input inference). (2) Implementing a multi-turn revision strategy to refine reasoning chains.

**Other Comments Or Suggestions:**

Please see the detailed feedback in QA section.

**Other Strengths And Weaknesses:**

Strengths:

* Innovative combination of structured code training with natural language reasoning tasks.
* Utilization of dual tasks (input inference and output prediction) that tap into the logic of executable functions.
* A comprehensive experimental evaluation that explores different prompt and revision strategies.
* The paper is easy to follow.

Weaknesses:

* Potential bias due to the selection criteria for code examples, which may not capture the full diversity of reasoning patterns.
* The multi-turn revision mechanism shows diminishing returns, and error propagation remains a concern.
* Key concepts (e.g., the deterministic reverse function) are insufficiently defined and justified, which may undermine the generality of the approach.


Please see the detailed feedback in QA section.

**Questions For Authors:**

I think this paper should be considered as a good attempt for combining the structured format with the natural language reasoning tasks. However, I may doubt that the diverse reasoning patterns cannot be fully captured by code format as some reasoning cases don’t closely mirror procedural code logic like theorem proof. Another concern is that although the authors aggregate code from multiple sources, the selection criteria (e.g., filtering for complexity) might bias the dataset toward certain types of reasoning or programming styles, potentially limiting the model’s exposure to broader reasoning scenarios. To be much clearer, recasting code execution as natural language input-output prediction may oversimplify or misrepresent non-procedural reasoning. By focusing on code-centric logic, the approach might neglect abstract, non-linear reasoning patterns inherent in broader tasks, questioning the generality of the improvements.

Another concern is regarding the multi-turn revision: although the multi-turn revision is intended to correct initial errors, the reported gains diminish after the first turn. This suggests that added complexity may not translate into substantial performance improvements, and the error propagation risks remain insufficiently addressed.

In Table 3, I’m curious about the combination of using query+code in the prompt and letting the response be both CoT+Code? Since it seems like these two setups have both promising results.

I have some concerns regarding the collection of the training data, especially for the process that all the responses for input-output pairs are generated by Deepseek-v2.5. The Figure 1 mentioned that the CoTs can have some optional revisions to further enhance the reasoning chains. I want to see more evidence for this claim, as I think the quality of the responses are vital for the whole system; several specific concerns could be (1) how to ensure that the generated responses contain the valid logic flow from the user query, (2) how to balance the complexity of the generated CoTs.

I’m confused with the statement from Line 167 to 178, what do you mean by the deterministic reverse function? Why is it important?

I’m curious compared to Code/I/O, how many instances can be fixed in Code/I/O++? And the results from Table 1 (w/ Code/I/O and w/ Code/I/O++) are quite similar and even in some setups using Code/I/O++ even results in worse performance; does it mean that even with additional verification and regeneration, there are still some errors that cannot be fixed.

**Relation To Broader Scientific Literature:**

I think this paper contributes to the intersection of code generation, and Cot-like reasoning. It builds on recent successes in using structured training data (from code and math tasks) to enhance reasoning in language models and connects with work on chain-of-thought prompting and iterative refinement. However, it could benefit from a deeper discussion of how its approach compares to methods that tackle abstract, non-linear reasoning, and why code-centric logic might limit exposure to broader reasoning patterns.

**Theoretical Claims:**

I don't think the paper give some theoretical claims as it seems to follow the typical SFT process.

---

> ### Author Rebuttal · Authors · 2025-04-01
>
> Thank you for the time and effort you spent reviewing our paper, and for recognizing our contributions. Below are our responses:
> # Q1
> > Diverse reasoning patterns may not be fully captured by code & Compare to other methods that tackle abstract, non-linear reasoning
>
> Thanks for this comment. We agree that not every reasoning pattern exists in code. However, as shown in case study (response to reviewer z6HH Q3), many foundational patterns can indeed be identified. We also tested on domains that are rare in code, e.g., medical reasoning (response to reviewer qybs Q7), and observed gains, indicating that the improvement is generalizable to some extent. Regarding comparison with other methods that tackle abstract, non-linear reasoning, we provide a discussion about neuro-symbolic systems (response to reviewer qybs Q4).
> # Q2
> > Code-centric logic might limit exposure to broader reasoning patterns: the selection criteria (e.g., filtering for complexity) might bias the dataset
>
> Thanks for this comment. Actually, the selection criteria are set for mitigating bias rather than enhancing it. In the CodeMix source, most samples are pure algorithms, thus we deliberately filtered out pure algorithms in PyEdu source and did not apply complexity-based filtering. However, we also acknowledge that certain biases may still exist, as we only include executable code with proper JSON input/output formats. We leave this issue as future work to explore.
> # Q3
> > Concerns on error propagation in multi-turn revision and discussion on recent advances in related topics.
>
> Thanks for this comment. We listed the statistics about multi-turn revision in Fig 7, Appendix D of the submission. Actually, only a small number of errors (16% in input pred and 10.7% in output pred) can be fixed. The value can be even smaller if a next round of revision is conducted.
>
> As a result, a slight improvement when involving this revision is expected. Potential reasons may be that DeepSeek-V2.5 still lacks the ability to revise. We did not tune this part with huge effort, as we only wanted to try this as a preliminary attempt to utilize the verifiable nature of code.
>
> On the other hand, there are also directions we could integrate into our workflow to enhance multi-turn revision effects. For example, involving multi-agent debate and discussion [1], interactive critiquing with diverse tools [2], or using models with strong self-reflection abilities [3]. We also plan to explore them in our future work to address error propagation more effectively.
>
> [1] Improving Factuality and Reasoning in Language Models through Multiagent Debate; ICML 2024
>
> [2] CRITIC: Large Language Models Can Self-Correct with Tool-Interactive Critiquing; ICLR 2024
>
> [3] DeepSeek-R1: Incentivizing Reasoning Capability in LLMs via Reinforcement Learning; arXiv:2501.12948
> # Q4
> >  The combination of using query+code in the prompt and letting the response be both CoT+Code in Table 3
>
> Thank you for this question. The two Code parts in prompt and response are actually identical - both refer to the reference code(see Table 8 for an example). If we include Code in both parts, the model will just learn to copy a block of contents, which would bring no benefits in learning. Therefore, we did not include this variant in the submission.
> # Q5
> > Quality concerns on the generated responses in CodeI/O: 1) if they contain the valid logic flow from the user query, 2) how to balance the complexity of the generated CoTs
>
> Thank you for this comment. Our observations show that responses with correct predictions usually demonstrate valid logical flow. While removing incorrect responses seems intuitive for improving quality, our results (Table 2, rows 1 and 5) show this actually degrades performance. We hypothesize that such filtering, though ensuring valid logic, reduces exposure to diverse reasoning patterns, particularly difficult ones. Besides, our two-stage training approach helps mitigate these flaws, as high-quality second-stage data can remediate first-stage issues.
>
> Regarding CoT complexity balancing, we find that CoT complexity naturally corresponds to the input/output prediction complexity. Our sampling across data sources implicitly covers different complexity levels, though we didn't explicitly balance this in a fine-grained manner, which we agree is an important direction and we leave it as future work to explore.
> # Q6
> > What is the deterministic reverse function (Line 167 to 178) and why is it important?
>
> Reverse functions take an output and return a feasible input for the original functions. If they are deterministic, one output maps to exactly one input, and we can just use the execution trajectory (i.e., print the executed lines of codes and the intermediate variables sequentially) as the perfect responses. However, since multiple inputs often produce the same output, truly deterministic reverse functions rarely exist. This is partly why we use DeepSeek-V2.5 to generate responses directly.

---

> > ### Comment · Reviewer_rx1K · 2025-04-03
> >
> > Thank you for the detailed, easy-to-follow responses. I have read them and still hold a positive opinion about this paper. Thanks!

---

> > > ### Author Response · Authors · 2025-04-05
> > >
> > > Thanks for your kind words in the response and your positive feedback on our paper once again!

---

### Official Review · Reviewer_qybs · 2025-03-11

**Overall Recommendation:** 4

**Summary:**

The paper introduces CODEI/O, a novel approach designed to enhance the reasoning capabilities of large language models by leveraging code input-output prediction. The key idea is to transform code into a format where models predict inputs or outputs given a function, while reasoning in natural language using COT rationales. The methodology aims to apply models to fundamental reasoning patterns, such as logic flow planning, state-space searching, and modular decomposition, without being constrained by code-specific syntax. The paper further enhances this method with CODEI/O++, which incorporates multi-turn revisions to refine CoTs. Experiments across multiple reasoning benchmarks demonstrate that CODEI/O and CODEI/O++ improve performance not only on code-related tasks but also on more general reasoning challenges, achieving more balanced and generalizable results than existing datasets.

**Claims And Evidence:**

Most of the claims are well evaluated.  One issue is that,  the paper claims that CODEI/O systematically condenses reasoning patterns embedded in code and enhances reasoning capabilities across various domains. "For the claim, "We believe that real-world code programs reflect the integration of a wide range of reasoning patterns across diverse contexts, making them an ideal source for training while minimizing the risk of overfitting". The paper should give more evidence to the claims. For example, some case studies or key observations could be provided to explain what similar patterns they share.

**Essential References Not Discussed:**

One potential gap is a comparison with neuro-symbolic integration methods, which also attempt to abstract structured reasoning. May add some discussions and comparisons on it.

**Experimental Designs Or Analyses:**

Yes. The experiments are well-designed. However, it lacks some qualitative analysis to show why the CODEI/O dataset is more effective than others. Moreover, the potential data leak risks from the the collected code should be discussed.

**Methods And Evaluation Criteria:**

Yes. The methods make sense for the enhancing the reasoning capabilities.  The experimental setup, including the comprehensive benchmarks and a comparison against strong baselines such as OpenMathInstruct, WebInstruct, and OpenCoder-SFT, is robust and provides clear evidence of the effectiveness of CODEI/O.

**Other Comments Or Suggestions:**

N.A.

**Other Strengths And Weaknesses:**

The approach is novel in that it reformulates reasoning as a structured code input-output prediction task with natural language CoTs, a departure from standard pre-training or direct instruction tuning. The work has strong implications for improving LLMs’ general reasoning ability beyond just code-related tasks.


Potential Weaknesses:

- While the method is well-validated experimentally, there is limited discussion on potential biases and data leaks in the collected dataset.
- Further insights into model interpretability and why CODEI/O leads to improvements across different reasoning tasks would be valuable.
- The training efforts should be clearly provided such as the number of GPUs used and the training time.

**Questions For Authors:**

1. How does the performance of CODEI/O compare when tested on completely unseen reasoning categories that may not align with the selected code dataset (e.g., medical domain or others)?

2. Have you analyzed whether the model is truly learning structured reasoning patterns, or if it is simply memorizing input-output mappings? Can you provide qualitative examples of reasoning improvements or the potential patterns?

3. How does CODEI/O perform when fine-tuned on smaller models (e.g., 1.5B parameters)?

4. Could you provide the training cost details?

5. How do you handle the potential bias or data leaks in the collected code dataset?

**Relation To Broader Scientific Literature:**

The paper positions CODEI/O as a bridge between code reasoning and broader natural language reasoning, distinguishing itself from prior works that focus purely on code execution (e.g., CRUXEval) or task-specific data augmentation. It builds upon previous work on structured reasoning datasets but extends the concept by systematically curating input-output mappings from code to enhance general model training.The authors reference relevant studies on code reasoning, execution-based learning, and dataset construction.

**Theoretical Claims:**

The paper does not provide theoretical claims.

---

> ### Author Rebuttal · Authors · 2025-04-01
>
> Thanks for the time and effort you spent reviewing our paper, and for recognizing our contributions. Below are our responses:
>
> # Q1
> > Evidence to show code integrates diverse reasoning patterns
>
> We conduct a case study on the CodeI/O dataset, and witness typical examples with certain reasoning patterns. Please refer to our response to Q3 of Reviewer z6HH for more details.
> # Q2
> > Qualitative analysis to show why the CODEI/O dataset is more effective than others
>
> We compare samples from CodeI/O and other baseline datasets. Key differences are:
>
> 1) Structured Reasoning Process: CodeI/O shows a standard problem-solving method with clear steps, while other datasets show more varied and less ordered reasoning.
>
> 2) Algorithmic Thinking Pattern: CodeI/O shows orderly step-by-step processes, unlike the more direct shortcuts in other datasets.
>
> 3) Complete Reasoning Traces: CodeI/O gives full traces that record the whole reasoning process, making it better for training models to explain their thinking.
>
> These differences stem from both the input/output prediction task and our DeepSeek-V2.5 response generation. We will add these analyses to our next paper version.
> # Q3
> > Data leak risks
>
> We conduct a strict 13-gram-based leakage detection on CodeI/O data following [1], the results are as follows:
>
> |Benchmark|Leakage Ratio (%)|
> |-|-|
> |LeetCode-O|21.5(950/4414)|
> |KorBench|5.1(64/1250)|
> |MATH/MMLU/CRUXEval|0.1|
> |Others|0|
>
> We see most of the benchmarks are not leaked. Upon manual inspection on the two ones with high ratio:
>
> - KorBench overlaps only contain general descriptions like Sudoku rules or common letter sequences ("A B C D...") rather than specific questions – our training tasks and the benchmark tasks are completely different.
>
> - LeetCode-O overlaps stem from sibling problems sharing common descriptions (e.g., Two Sum I & II), despite that we have removed all original problems from our training data
>
> To further detect whether the gains on these two are due to data leakage, we calculated the sample-wise acc gains of CodeIO compared to the baseline on both the full set (F) and the non-leaked set (UN). The results are as follows:
>
> ||LeetCode-O||KorBench||
> |-|-|-|-|-|
> ||F|UN|F|UN|
> |Qwen|3.8|3.9|5.8|6.1|
> |LLaMA|9.4|9.4|1.0|0.9|
> |DSLite|5.3|5.7|-1.2|-1.3|
> |Gemma|3.7|3.9|1.4|1.3|
>
> The similar gains on both full and unleaked subsets across all models confirm that our improvements are not affected by data leakage.
>
> These analyses will be in our next paper version.
>
> [1] Evaluation data contamination in LLMs: how do we measure it and (when) does it matter? arXiv:2411.03923
> # Q4
> > Discussions on neuro-symbolic methods
>
> Thanks for this comment. Our work can also be regarded as neuro-symbolic integration, specifically in the "neuro:symbolic->neuro" category per [1]. We use symbolic rules (Python execution) to guide LLM training, but rely solely on neural components in inference. This mainly differs from other categories like Symbolic[Neuro], Neuro[Symbolic], or Neuro|Symbolic, which utilizes both techniques in inference. Further discussion will be included in the next revision.
>
> [1] Towards Cognitive AI Systems: a Survey and Prospective on Neuro-Symbolic AI, arXiv 2401.01040
> # Q5
> > Further insights into interpretability & Qualitative examples of reasoning improvements
>
> Thanks for this advice. We conduct a case study on model behavior in response to Q3 of Reviewer z6HH. Please kindly refer to that for details. Also, as the input-output prediction accuracy is only about 50% in CodeI/O data, it’s hard to memorize them and hack the benchmarks, and the gains should mostly come from the underlying reasoning logic flow.
> # Q6
> > Training costs
>
> |Model|# of GPUs (40GB A100)|Stage1(CodeI/O) (hrs)|Stage1(CodeI/O++) (hrs)|Stage2 (hrs)|
> |-|-|-|-|-|
> |Qwen|80|5.8|7.5|3.5|
> |LLaMA|80|6.15|8.4|4.0|
> |DSLite|80|4.4|7.0|2.5|
> |Gemma|160|14.0|18.5|7.5|
> # Q7
> > Performance on unseen reasoning categories
>
> We test on two medical reasoning tasks for complex clinical diagnosis: MedQA (US subset) [1] and MedBullets [2]. The results on Qwen 7B 2.5 Coder are as follows, indicating CodeI/O can also improve categories that may not align with codes:
>
> |Data|MedQA|MedBullets|
> |-|-|-|
> |Stage2 Only|47.2|40.9|
> |CodeI/O|49.3|42.5|
> |CodeI/O++|49.2|42.5|
> |OMI2|48.1|39.9|
> |WI|47.8|40.3|
> |PyEdu|46.2|42.5|
> |OC-SFT-1|48.0|38.6|
>
> [1] What disease does this patient have? a large-scale open domain question answering dataset from medical exams; arXiv:2009.13081
>
> [2] Benchmarking Large Language Models on Answering and Explaining Challenging Medical Questions; arxiv 2402.18060
> # Q8
> > Performance on smaller models
>
> We test Qwen 2.5 1.5B as follows:
>
> |Data|Avg|
> |-|-|
> |Stage2 Only|37.4|
> |CodeI/O|38.3|
> |CodeI/O++|38.5|
> |OMI2|37.7|
> |WI|37.1|
> |PyEdu|37.8|
> |OC-SFT-1|37.4|
>
> The results show CodeIO is still effective, though with less gains than in larger models. This suggests smaller models may lack sufficient capacity to fully leverage the reasoning patterns in our datasets.

---

### Official Review · Reviewer_bnUn · 2025-03-14

**Overall Recommendation:** 4

**Summary:**

In this work, the authors develop CodeI/O and CodeI/O++ to improve the reasoning capabilities of Large Language Models. The proposed CodeI/O approach trains models to predict code inputs and outputs in natural language. Evaluation on a variety of benchmark datasets and base models show that CodeI/O improves reasoning capabilities even in non-code domains. Additionally, the authors provide extensive ablation studies to justify the design choices when building CodeI/O and CodeI/O++.

**Claims And Evidence:**

The main claims in the paper are all supported by extensive experimental evidence across a variety of benchmark tasks and base models. Additionally, all of the claims about building the finetuning dataset are supported by extensive ablation studies. One minor point is that that the authors claim "CodeI/O and CodeI/O++ exhibit universal effectiveness across model sizes and architectures," which is not exactly true given that the proposed method does underperform on some base models/tasks.

**Essential References Not Discussed:**

I am not familiar enough with this research area to be able to comment on any missing literature.

**Experimental Designs Or Analyses:**

All of the experiments seemed well designed to me, but I am less familiar with this research area and may not be aware of any weakness/flaws. In my view, the main studies are well designed and provide a fair evaluation, and the ablation studies provide important insight into the design of the finetuning datasets and the scaling of the proposed method.

**Methods And Evaluation Criteria:**

I will say that I am less familiar with this research area, but the selection of benchmark datasets and base models seems reasonable to me. Evaluating across multiple different base models provides stronger support for the proposed method and evaluating on tasks outside of just code reasoning demonstrates that CodeI/O provides improvements to general reasoning performance.

**Other Comments Or Suggestions:**

### Update After Rebuttal

The authors provided some additional experiments which show the method still works for larger models. Overall this is a strong work and I recommend acceptance of the paper.

**Other Strengths And Weaknesses:**

Strengths:
- S1: CodeI/O and CodeI/O++ provide consistent performance improvements across a variety of benchmark tasks and base models
- S2: The authors provide several strong ablations studies that demonstrate impressive performance scaling and provide insights into the design of the finetuning dataset.
- S3: The paper is well written, easy to understand, and has nice figures.

Weaknesses:
- W1: Most of the base models used in the paper are relatively small and the proposed method has the smallest improvement in performance on Gemma 2 27B. As such, it is unclear if the results demonstrated in the paper will scale to larger models.

**Questions For Authors:**

- Q1: Can the authors attempt to evaluate the proposed method with a larger base model?

**Relation To Broader Scientific Literature:**

Reasoning capabilities are incredibly important to continue to improve LLM performance, and the proposed method provides a strong way to improve LLM reasoning performance across domains.

**Theoretical Claims:**

N/A -- no theoretical claims are made.

---

> ### Author Rebuttal · Authors · 2025-04-01
>
> Thank you for the time and effort you spent reviewing our paper, and for recognizing our contributions. Below, we have listed our responses to your questions and comments.
> # Q1
> > The claim "CodeI/O and CodeI/O++ exhibit universal effectiveness …" is not exactly true given that the proposed method does underperform on some base models/tasks.
>
> Thanks for this comment, we will change this statement to a more precise one, e.g., “CodeI/O and CodeI/O++ demonstrate performance improvements across models of various sizes and architectures on most benchmarks, although we also observed nearly unchanged or even decreased performance on a small number of tasks.”
> # Q2
> >  Can the authors attempt to evaluate the proposed method with a larger base model?
>
> Thanks for this suggestion. We have trained a larger model, LLaMA3 70B. The results are as follows:
>
> | | Wino-Grande | DROP | GSM8K | MATH | GPQA | MMLU-STEM | LC-O | CRUX-I | CRUX-O | BBH | BBH-ZH | Zebra-Logic | Kor-Bench | Live-Bench | AVG |
> |-|-|-|-|-|-|-|-|-|-|-|-|-|-|-|-|
> | Baseline (Instruction Tuning) | 75.9 | **86.5** | **93.6** | 68.9 | 44.2 | 85.8 | 24.8 | 65.0 | 72.6 | 83.9 | 88.6 | 20.0 | 51.8 | 42.6 | 65.6 |
> | CodeI/O + Instruction Tuning | **76.3** | 85.8 | 93.1 | **70.1** | **45.3** | **85.9** | **31.4** | **68.8** | **76.1** | **85.9** | **88.7** | **22.6** | **53.8** | **44.1** | **67.4** |
>
> The results show that CodeI/O also works well on larger models. Although we see some performance drop on a small set of benchmarks, gains are obtained on most of them, indicating an overall improvement in reasoning ability.

---

> > ### Comment · Reviewer_bnUn · 2025-04-01
> >
> > I thank the authors for providing this additional experiment with a larger model. This is a good result to include in the revised manuscript, however it does not change my evaluation and I will maintain my score.

---

> > > ### Author Response · Authors · 2025-04-05
> > >
> > > Thank you for recognizing the additional experimental results we provided, and we appreciate your positive feedback on our paper once again!

---

### Decision · Program_Chairs · 2025-05-01

**Decision:**

Accept (oral)

**Comment:**

The paper "CodeIO: Condensing Reasoning Patterns via Code Input-Output Prediction" presents a novel approach to enhance the reasoning capabilities of Large Language Models (LLMs) by leveraging code input-output prediction. The authors propose transforming code into a format where models predict inputs or outputs given a function, while reasoning in natural language using Chain-of-Thought (CoT) rationales. The methodology aims to apply models to fundamental reasoning patterns, such as logic flow planning, state-space searching, and modular decomposition, without being constrained by code-specific syntax. The paper further enhances this method with CodeI/O++, which incorporates multi-turn revisions to refine CoTs. Experiments across multiple reasoning benchmarks demonstrate that CodeI/O and CodeI/O++ improve performance not only on code-related tasks but also on more general reasoning challenges, achieving more balanced and generalizable results than existing datasets.

The reviewers are unanimous and recommend to accept. Although they suggest that the authors address the following points in the final version:
* Clarify Claims: The authors should clarify their claims regarding the "universal" effectiveness of the method and provide more precise language.
* Address Bias and Leakage: The authors should provide more detailed analyses of potential biases and data leakage in the dataset and discuss how these issues might be mitigated.

Nonetheless, the rebuttal period was productive and the paper is in really good shape for publication.

The method is interesting, relevant for all the code generation (broadly) LLMs, and its impact is thoroughly demonstrated, so I recommend for Oral.